# Plasma metabolomics reveals the shared and distinct metabolic disturbances associated with cardiovascular events in coronary artery disease

Jiali Lv[1,2,3,9], Chang Pan[1,2,4,5,9], Yuping Cai [6,7,9], Xinyue Han[1,2,3], Cheng Wang[3,8], Jingjing Ma[1,2,4,5], Jiaojiao Pang [1,2,4,5], Feng Xu [1,2,4,5], Shuo Wu [1,2,4,5], Tianzhang Kou[6], Fandong Ren[6], Zheng-Jiang Zhu[6,7] ✉, Tao Zhang [1,2,3] ✉, Jiali Wang [1,2,4,5] ✉ & Yuguo Chen [1,2,4,5] ✉

Risk prediction for subsequent cardiovascular events remains an unmet clinical issue in patients with coronary artery disease. We aimed to investigate prognostic metabolic biomarkers by considering both shared and distinct metabolic disturbance associated with the composite and individual cardiovascular events. Here, we conducted an untargeted metabolomics analysis for 333 incident cardiovascular events and 333 matched controls. The cardiovascular events were designated as cardiovascular death, myocardial infarction/stroke and heart failure. A total of 23 shared differential metabolites were associated with the composite of cardiovascular events. The majority were middle and long chain acylcarnitines. Distinct metabolic patterns for individual events were revealed, and glycerophospholipids alteration was specific to heart failure. Notably, the addition of metabolites to clinical markers significantly improved heart failure risk prediction. This study highlights the potential significance of plasma metabolites on tailed risk assessment of cardiovascular events, and strengthens the understanding of the heterogenic mechanisms across different events.

Coronary artery disease (CAD) remains the leading cause of mortality and morbidity worldwide, despite enormous advances in the therapeutic strategies over the past 40 years[1]. The lingering incidences of cardiovascular events, including cardiovascular death, myocardial infarction (MI) and stroke, and heart failure (HF), impose significant socioeconomic burdens and negatively impact the quality of patients' life[2,3]. It is urgent to precisely evaluate the risk of the cardiovascular complications and implement risk-guided secondary prevention[4,5]. However, the risk assessment geared toward the overall cardiovascular events is indefensible, since there may be

[1]Department of Emergency Medicine, Qilu Hospital of Shandong University, Jinan, China. [2]Shandong Provincial Clinical Research Center for Emergency and Critical Care Medicine, Qilu Hospital of Shandong University, Jinan, China. [3]Department of Biostatistics, School of Public Health, Cheeloo College of Medicine, Shandong University, Jinan, China. [4]Shandong Provincial Engineering Laboratory for Emergency and Critical Care Medicine, Qilu Hospital of Shandong University, Jinan, China. [5]Key Laboratory of Cardiovascular Remodeling and Function Research, Chinese Ministry of Education, Chinese National Health Commission and Chinese Academy of Medical Sciences, Qilu Hospital of Shandong University, Jinan, China. [6]Interdisciplinary Research Center on Biology and Chemistry, Shanghai Institute of Organic Chemistry, Chinese Academy of Sciences, Shanghai, China. [7]Shanghai Key Laboratory of Aging Studies, Shanghai, China. [8]National Institute of Health Data Science, Shandong University, Jinan, China. [9]These authors contributed equally: Jiali Lv, Chang Pan, Yuping Cai. ✉e-mail: jiangzhu@sioc.ac.ccn; taozhang@sdu.edu.cn; wangjiali_2000@126.com; chen919085@sdu.edu.cn

great heterogeneity in the progression and occurrence of these complications initiated from CAD. Clarifying the shared and distinct metabolic disturbance has the potential to identify biomarkers for tailored risk assessment of future cardiovascular events, in addition to improve our understanding of the molecular processes involved in different outcomes.

Several circulating biomarkers such as cardiac troponin T/I (cTn T/I) and N-terminal pro-brain natriuretic peptide (NT-proBNP) have been identified as predictors of adverse cardiovascular events[6,7]. However, these established biomarkers are usually detected after myocardial injury or prominent cardiac dysfunction[8,9]. Earlier detection of plasma biomarkers and identification of molecules especially associated with cardiovascular events is of significant importance for the prognostic evaluation and prompt intervention in patients with CAD. Previous studies have reported several metabolites that could predict the risk of cardiovascular events[10–12], however, these studies underestimated the specific biomarkers across different cardiovascular outcomes. Therefore, it is necessary to identify the metabolic profiles towards the shared and distinct metabolic disturbance associated with the cardiovascular events.

Metabolomics is a promising and powerful way to reveal metabolic disorders associated with pathological conditions[13,14]. Compared to genome and proteome studies, metabolomics emerges as the closest omics layer that relates to the phenome[15]. In light of the substantial heterogeneity in the mechanisms of individual cardiovascular events initiated from CAD, the abundant information behind metabolomic profiles has been insufficiently considered in the risk prediction of cardiovascular events. Untargeted metabolomics enables the global detection and relative quantification of metabolites, allowing for a comprehensive understanding of metabolic changes during disease progression[14–16], which provides a promising approach to extensively discover circulating metabolite biomarkers for risk prediction in patients with CAD.

In this work, we conduct a case-control study nested within the BIomarker-based Prognostic assessment for patients with stable angina and acute coronary syndrome (BIPass) cohort[17]. We investigate the shared plasma metabolic profile in patients with the composite of cardiovascular events, compared to those without events. Acylcarnitine dysregulation serves as the shared metabolic profile for the composite of cardiovascular events, including cardiovascular death, HF, and MI/stroke. HF exhibits distinct metabolic disturbance characterized by disturbed glycerophospholipids metabolism, which differs from other cardiovascular events. We further identify distinct metabolite biomarkers and pathways specific to individual cardiovascular events. The inclusion of metabolites into the Thrombolysis In Myocardial Infarction (TIMI) variables improves the predictive capability for the composite of cardiovascular events. Notably, the addition of metabolites to TIMI variables, high sensitivity cardiac troponin T (hs-cTnT), and NT-proBNP causes significant enhancement in HF risk prediction. Altogether, these findings reveal the shared and distinct metabolic disturbance has the potentiality to identify biomarkers for risk assessment of cardiovascular events, as well as improve our understanding of the molecular processes involved in different outcomes.

## Results
### Characteristics of study participants
In this study, a total of 666 CAD patients were included for this untargeted metabolomics study, involving 333 patients with incident cardiovascular events (case group) and 333 patients without any events (control group) (Fig. 1). The 666 CAD patients had a median age of 68.2 years (interquartile range 61.7–74.8 years), 255 (38.3%) were female, the median body-mass index was 24.8 kg/m² (interquartile range 22.9–27.2 kg/m²), the median systolic blood pressure was 134 mmHg (interquartile range 122–150 mmHg), and the median low-

density lipoprotein-cholesterol level was 2.2 mmol/L (interquartile range 1.8–2.8 mmol/L); 19.7% of patients were current smokers, and 44.4% had diabetes. Patients with the composite of cardiovascular events demonstrated higher levels of heart rates, white blood cell counts, creatinine, cystatin C, hs-cTnT, and NT-proBNP, but lower levels of estimated glomerular filtration rate (eGFR), and hemoglobin, compared to those without any cardiovascular events (Table 1). The further baseline patient characteristics in the discovery and validation set, and stratified by individual cardiovascular events (cardiovascular death, HF, and MI/stroke) are presented in Table 1, and Supplementary Data 1–3. The medications during follow-up are presented in the Supplementary Table 1.

### Shared metabolic profile for cardiovascular events
Patients with the composite of cardiovascular events displayed a clearly distinguishable metabolic profile compared to those in the control group (Fig. 2a). Metabolic pathway analysis unveiled 19 dysregulated metabolic pathways related to the composite of cardiovascular events. These pathways encompass tyrosine metabolism, cysteine, and methionine metabolism, pentose and glucuronate interconversions, lysine degradation, and fatty acid biosynthesis (Supplementary Data 4). We identified 82 metabolites as differential metabolites for predicting the risk of the composite of cardiovascular events (Fig. 2b). After adjusting for the TIMI variables, NT-proBNP and hs-cTnT, a total of 23 differential metabolites were significantly associated with the risk of the composite of cardiovascular events (Fig. 2c).

Among these 23 metabolites, it was notable that the large proportion was middle and long-chain acylcarnitines (13 ≤ carbon number ≤ 25). Nine acylcarnitines showed significantly higher levels in patients with cardiovascular events than those in controls. Plasma phthalide and 5-Acetylamino-6-amino-3-methyluracil (AAMU) showed decreased levels, while the remaining metabolites showed increased levels in patients with the composite of cardiovascular events, compared to those in controls. Furthermore, we identified the combination of shared key differential metabolites for predicting the risk of overall cardiovascular events, which could serve as potential biomarkers (Supplementary Data 5). The combination of key metabolites was composed of 14 metabolites, including 12 up-regulated and 2 down-regulated metabolites (Supplementary Fig. 1).

### Distinct metabolic patterns for individual cardiovascular events
We applied the same workflow which was used for the selection of differential metabolites related to the composite of cardiovascular events to identify metabolic biomarkers specific to each individual cardiovascular event (Fig. 3a). The metabolic profile between each individual cardiovascular event and the control group showed distinct tendency of separation (Supplementary Fig. 2). Initial analyses revealed 94 metabolites related to cardiovascular death, 121 metabolites related to HF, and 63 metabolites related to MI/stroke, respectively (Supplementary Data 6–8, Supplementary Fig. 3). To further investigate the predictive capability of these metabolites, we identified the combination of specific key differential metabolites for predicting each type of cardiovascular event (Supplementary Figs. 4–6).

The metabolites significantly associated with cardiovascular death, HF, and MI/stroke included 25 carnitines, 3 fatty acyls, 1 steroid and steroid derivative, and 11 metabolites of other/unknown classes (Fig. 3b, c). Patients with HF showed a specific metabolic alteration characterized by metabolic dysregulation of glycerophospholipids, which was differed from those with cardiovascular death and MI/stroke (Fig. 3b). These findings suggest that HF possesses a unique risk profile compared to cardiovascular death and MI/stroke, and disturbed lipid metabolism may play a pivotal role in the pathophysiological processes of HF.

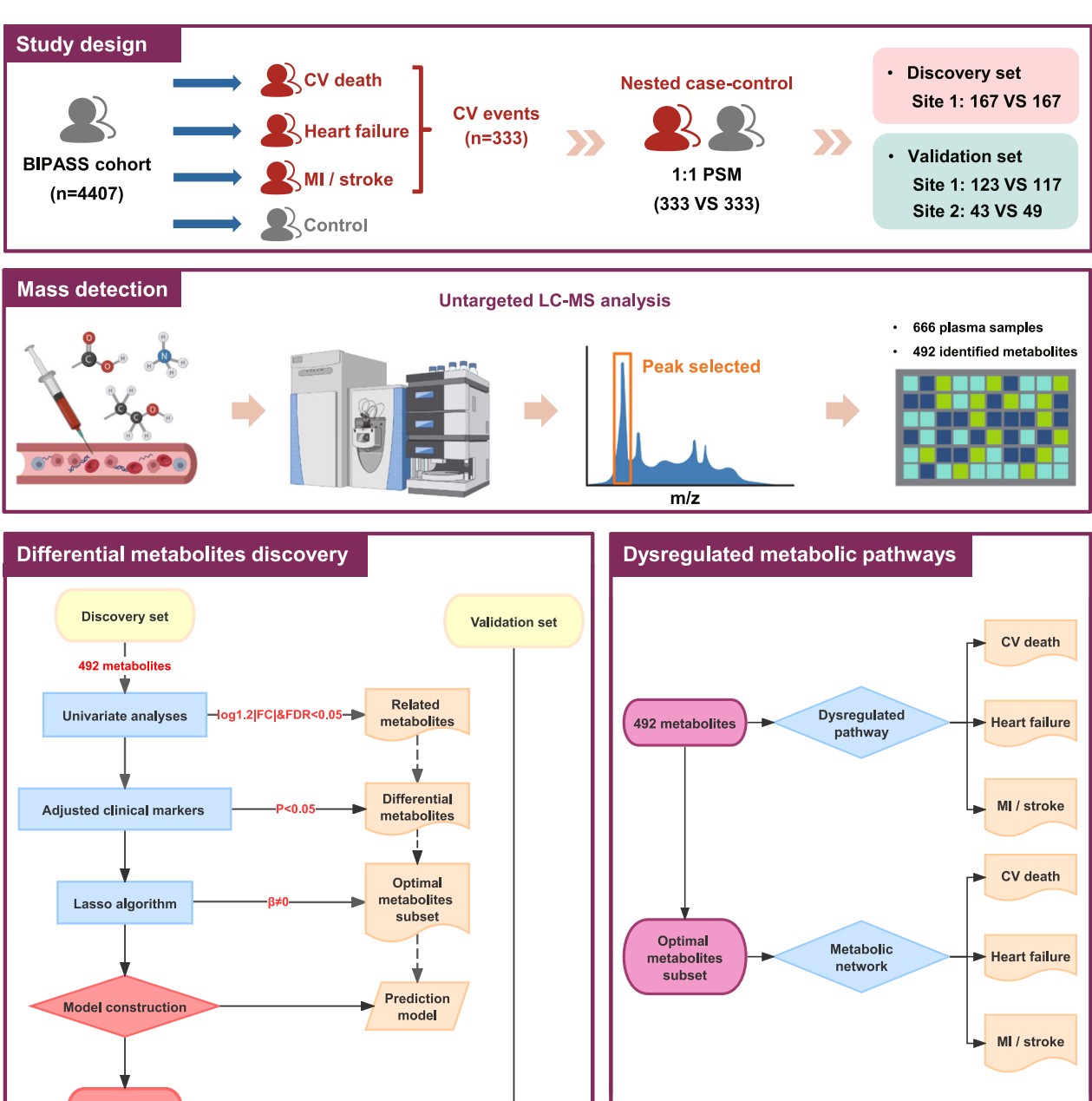

**Fig. 1 | Flowchart of the untargeted metabolomics study.** BIPass BIomarker-based Prognostic assessment for patients with stable angina and acute coronary syndrome; CV cardiovascular; FC fold change; FDR false discovery rate; Lasso the least absolute shrinkage and selection operator; LC-MS the liquid chromatography-mass spectrometry; MI myocardial infarction; PSM propensity score matching.

## Metabolic networks and pathways related to cardiovascular events

To gain insights into the molecular mechanisms of metabolites in the development of cardiovascular events, we constructed event-specific metabolic networks based on the combination of key differential metabolites for cardiovascular death, HF, and MI/stroke, respectively (Fig. 3d–f, Supplementary Table 2). In the metabolic network associated with cardiovascular death, oxohexadecadienoylcarnitine (Car (18:3-O)) and tetradecadienoylcarnitine (Car (14:2)) emerged as crucial metabolites, displaying a positive correlation ($r = 0.481$). For the metabolic network of HF, a cluster of glycerophospholipids, phosphatidylcholine (PC) (8:0/8:0), PC (10:0/10:0), lysophosphatidylcholine (LPC) (O-16:0) and lysophosphatidic acid (20:5), played a pivotal role in the complicated interaction of metabolites. Furthermore, PC

(22:2) and LPC (20:4) exhibited a strong positive correlation within the HF network ($r = 0.589$). In the metabolic network specific to MI/stroke, we found carnitine cluster was the predominant element, including palmitoleylcarnitine (Car (16:1)), hexadecatrienoylcarnitine (Car (16:3)), and oxooctanoylcarnitine (Car (8:1-O)).

Pathway analyses unveiled that the individual cardiovascular events mainly shared amino acid and lipids-related pathways, including cysteine and methionine metabolism, tyrosine metabolism, tryptophan metabolism, phenylalanine metabolism, fatty acid biosynthesis, and propanoate metabolism (Fig. 3g, Supplementary Data 9–11). The cardiovascular death and HF had the shared dysregulated metabolic pathways, including lysine degradation, glycerophospholipid metabolism, steroid hormone biosynthesis, galactose metabolism, and glutathione metabolism. The common metabolic

**Table 1 | Clinical characteristics of the study participants**

| Variable | Discovery set | | Validation set | |
|---|---|---|---|---|
| | Case (*n* = 167) | Control (*n* = 167) | Case (*n* = 166) | Control (*n* = 166) |
| **Demographics** | | | | |
| Age, years | 70.1 [63.9, 77.0] | 68.0 [61.5, 74.3] | 65.9 [60.3, 73.6] | 68.4 [62.2, 74.7] |
| Body mass index, kg/m$^2$ | 24.5 [22.2, 27.2] | 24.9 [22.9, 27.0] | 25.0 [23.2, 27.4] | 24.8 [22.7, 27.1] |
| Males, *n* (%) | 107 (64.1) | 100 (59.9) | 105 (63.3) | 99 (59.6) |
| **Risk factors and medical history** | | | | |
| Current smokers, *n* (%) | 34 (20.4) | 30 (18.0) | 35 (21.1) | 32 (19.3) |
| Hypertension, *n* (%) | 122 (73.1) | 119 (71.3) | 117 (70.5) | 125 (75.3) |
| Diabetes mellitus, *n* (%) | 65 (38.9) | 75 (44.9) | 77 (46.4) | 79 (47.6) |
| Congestive heart failure, *n* (%) | 5 (3.0) | 1 (0.6) | 3 (1.8) | 0 (0.0) |
| Peripheral arterial disease, *n* (%) | 7 (4.2) | 4 (2.4) | 3 (1.8) | 5 (3.0) |
| Previous myocardial infarction, *n* (%) | 32 (19.2) | 33 (19.8) | 38 (22.9) | 29 (17.5) |
| Previous stroke, *n* (%) | 31 (18.6) | 22 (13.2) | 36 (21.7) | 27 (16.3) |
| Previous PCI/CABG, *n* (%) | 36 (21.6) | 39 (23.4) | 39 (23.5) | 35 (21.1) |
| eGFR, mL/min/1.73 m$^2$ | 90.2 [71.6, 111.3] | 100.4 [84.7, 117.0] | 93.7 [73.6, 113.5] | 99.4 [82.2, 119.9] |
| Coronary stenosis (≥50%), *n* (%) | 54 (32.3) | 48 (28.7) | 52 (31.3) | 47 (28.3) |
| Echo left ventricular ejection fractions | 45.0 [35.0, 60.0] | 60.0 [48.8, 66.0] | 56.0 [37.5, 62.0] | 61.0 [55.0, 65.0] |
| **Pre-hospital medications** | | | | |
| β receptor blockers, *n* (%) | 55 (32.9) | 66 (39.5) | 66 (40.7) | 63 (39.6) |
| ACEI/ARB, *n* (%) | 40 (24.2) | 43 (26.1) | 44 (27.0) | 54 (34.2) |
| Statins, *n* (%) | 77 (46.1) | 84 (50.6) | 85 (52.5) | 94 (59.9) |
| Aspirin, *n* (%) | 111 (67.3) | 113 (68.1) | 119 (73.0) | 123 (76.9) |
| **Admission characteristics** | | | | |
| Heart rate, beats/min | 75.0 [63.0, 87.5] | 72.0 [63.0, 80.0] | 77.0 [67.0, 84.0] | 70.0 [64.0, 78.0] |
| Systolic blood pressure, mmHg | 131.0 [117.5, 146.5] | 134.0 [118.0, 149.0] | 134.5 [123.0, 153.8] | 139.0 [127.0, 153.0] |
| **Admission diagnosis** | | | | |
| Stable angina, *n* (%) | 10 (6.0) | 19 (11.4) | 7 (4.2) | 16 (9.6) |
| Acute coronary syndrome, *n* (%) | 157 (94.0) | 148 (88.6) | 159 (95.8) | 150 (90.4) |
| **Biochemical analyses** | | | | |
| Hemoglobin, g/L | 129.0 [118.5, 142.0] | 135.0 [125.0, 144.0] | 130.0 [119.0, 145.0] | 133.0 [125.0, 143.0] |
| White blood cell, 10$^9$/L | 6.5 [5.6, 7.6] | 6.0 [5.0, 7.2] | 6.9 [5.6, 8.4] | 6.3 [5.4, 7.3] |
| LDL-C, mmol/L | 2.3 [1.8, 2.8] | 2.2 [1.7, 2.9] | 2.4 [1.8, 2.9] | 2.1 [1.8, 2.7] |
| HDL-C, mmol/L | 1.0 [0.8, 1.2] | 1.0 [0.9, 1.2] | 1.0 [0.8, 1.2] | 1.0 [0.9, 1.2] |
| Triglycerides, mmol/L | 1.3 [1.1, 1.8] | 1.3 [0.9, 1.7] | 1.4 [1.0, 2.1] | 1.4 [1.0, 1.9] |
| Creatinine, μmol/L | 77.0 [65.5, 93.5] | 71.0 [60.0, 85.0] | 75.5 [65.0, 92.8] | 70.0 [61.0, 83.0] |
| Cystatin C, mg/L | 1.0 [0.8, 1.3] | 0.9 [0.8, 1.1] | 1.0 [0.9, 1.3] | 1.0 [0.8, 1.2] |
| hs-cTnT, ng/L | 51.1 [13.9, 599.9] | 11.6 [6.6, 33.2] | 36.0 [12.9, 260.8] | 14.2 [6.9, 27.7] |
| NT-proBNP, ng/L | 7.2 [6.0, 8.2] | 5.9 [4.8, 6.7] | 6.6 [5.5, 7.8] | 5.5 [4.5, 6.6] |

Case indicates the composite of cardiovascular events. Control indicates no any cardiovascular events. Data are median with interquartile range, or *n* (%).

*ACEI* angiotensin converting enzyme inhibitors, *ARB* angiotensin receptor blocker, *CABG* coronary artery bypass grafting, *eGFR* estimated glomerular filtration rate, *HDL-C* high density lipoprotein-cholesterol, *hs-cTnT* high-sensitivity cardiac troponin T, *LDL-C* low density lipoprotein-cholesterol, *NT-proBNP* N-terminal pro-B-type natriuretic peptide, *PCI* percutaneous coronary intervention.

pathways for cardiovascular death and MI/stroke included the citrate cycle (TCA cycle) and glyoxylate and dicarboxylate metabolism, while HF and MI/stroke shared only the metabolic pathway of pentose and glucuronate interconversions. Amino acid metabolism predominated as the dysregulated metabolic pathways in HF. For cardiovascular death, the specific metabolic pathways were primarily enriched in starch and sucrose metabolism, nicotinate and nicotinamide metabolism. In contrast, the specific metabolic pathways for MI/stroke were composed of sphingolipid metabolism and linoleic acid metabolism.

### Association of metabolites with cardiovascular events risk
Figure 4 illustrates the associations between differential metabolites and cardiovascular events. The differential metabolites which were significantly associated with the risk of the composite and individual cardiovascular events both in the discovery set and validation set were presented in the forest plot, after adjusting for TIMI variables, NT-proBNP, and hs-cTnT. The association of N4-Acetylcytidine (OR: 1.55, 95% CI 1.15–2.12), 3-Methoxy-4-hydroxyphenylglycol sulfate (OR: 1.57, 95% CI 1.06–2.47), and N-[3-(2-oxopyrrolidin-1-yl)propyl] acetamide (OR: 1.40, 95% CI 1.08–1.83) with the composite of cardiovascular events risk remained significant in the validation set. Among them, N4-Acetylcytidine and 3-Methoxy-4-hydroxyphenylglycol sulfate, exhibited significantly positive association with the risk of cardiovascular death, with an OR of 1.57 (95% CI 1.02–2.27) and an OR of 1.85 (95% CI 1.12–3.43), respectively. Notably, these two differential metabolites exhibited stronger associations with HF risk, with an OR of 2.19 (95% CI 1.37–3.75) and an OR of 2.62 (95% CI 1.36–5.78), respectively. In

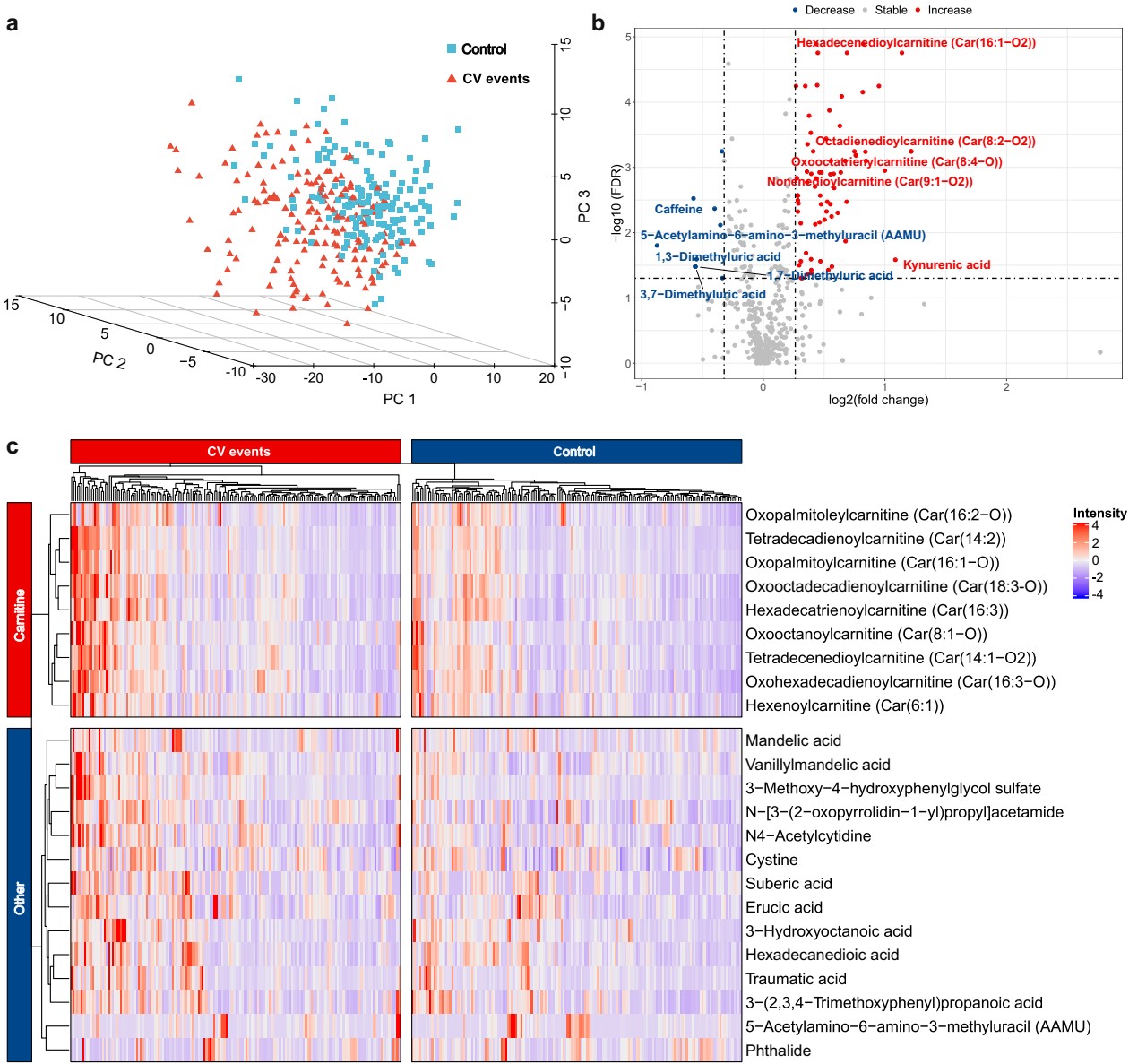

**Fig. 2 | Metabolic profile altered in patients with the composite of cardiovascular events. a** The partial least-squares discrimination analysis (PLS-DA) 3D score plots discriminated patients with the composite of cardiovascular events and the controls. **b** The volcano plot displayed the top five down-regulated and the top five up-regulated metabolites in the composite of cardiovascular events group. The red points represented the metabolites showing increased trend in the composite of cardiovascular events group, while the blue points represented the metabolites showing decreased trend in the composite of cardiovascular events group. The top five down-regulated and the top five up-regulated metabolites were further marked with their name. **c** The heatmap of differential metabolites related with the composite of cardiovascular events, after adjusting for TIMI variables, NT-proBNP and hs-cTnT. CV cardiovascular; hs-cTnT high sensitivity cardiac troponin T; NT-proBNP N-terminal pro-brain natriuretic peptide; PC principal component; TIMI Thrombolysis in myocardial infarction.

addition, PC (8:0/8:0), LPC (18:3), LPC (20:4), and erucic acid maintained significant associations with HF risk, with an OR of 0.52 (95% CI 0.27–0.93), 0.39 (95% CI 0.17–0.77), 1.79 (95% CI 1.09–3.07) and 1.67 (95% CI 1.08–2.70), respectively. Car (16:1) and N4-Acetylcytidine showed significant association with the risk of MI/stroke, with an OR of 1.42 (95% CI 1.01–2.02) and 1.50 (95% CI 1.07–2.16), respectively. The association of differential metabolites with the risk of the composite and individual cardiovascular events remains significant after further adjusting for pre-hospital medical treatments, the severity of coronary stenosis, and different clinical phenotypes (Supplementary Data 12).

### Predictive performance of the altered metabolites
To assess the ability of the combination of key metabolites to predict the risk of the composite and individual cardiovascular events, we calculated the area under curves (AUCs) of established risk factors (TIMI variables, hs-cTnT or NT-proBNP) with and without addition of the combination of key metabolites (Table 2). Notably, the addition of metabolites to the TIMI variables significantly increased the predictive performance for the composite of cardiovascular events (TIMI variables AUC: 0.63, 95% CI 0.57–0.69, TIMI variables and Metabolites AUC: 0.70, 95% CI 0.64–0.75), CV death (TIMI variables AUC: 0.68, 95% CI 0.60–0.74, TIMI variables and Metabolites AUC: 0.76, 95% CI 0.70–0.83) and HF (TIMI variables AUC: 0.68, 95% CI 0.59–0.77, TIMI variables and Metabolites AUC: 0.91, 95% CI 0.86–0.94). In particular, the addition of the combination of key metabolites to established risk factors significantly increased the predictive performance for HF risk: hs-cTnT AUC: 0.82, 95% CI 0.74–0.88, hs-cTnT and Metabolites AUC: 0.89, 95% CI 0.84–0.94, NT-proBNP AUC: 0.88, 95% CI 0.83–0.93, NT-

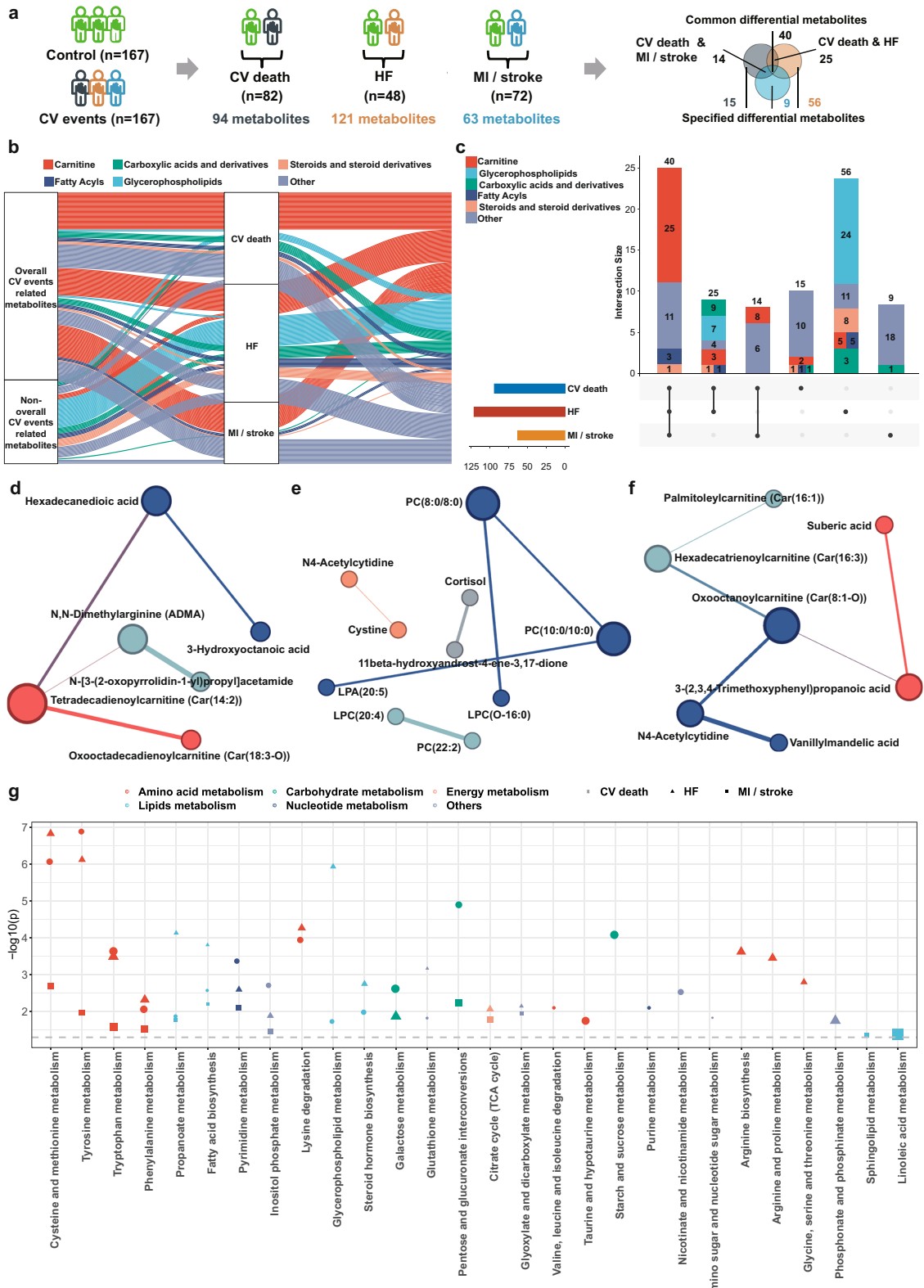

proBNP and Metabolites AUC: 0.94, 95% CI 0.91–0.97. Most prediction models exhibited good calibration, except for hs-cTnT (Supplementary Table 3, Supplementary Figs. 7–10). Incorporating metabolites into the hs-cTnT model, the calibration significantly improved.

The distribution of differential metabolites for related to the composite and individual cardiovascular events revealed consistent trends across different disease subtypes (Supplementary Figs. 11–14).

In sensitivity analysis, these findings were generally robust when including patients with acute coronary syndrome only (Supplementary Data 13 & Supplementary Table 4).

## Discussion

In this study, we had several important findings. (1) A total of 23 plasma metabolites were identified to be associated with the composite of

**Fig. 3 | Differential metabolites for individual cardiovascular events. a** The workflow of differential metabolites analyses for individual cardiovascular events. There were 40 overlapping differential metabolites in the composite of cardiovascular events (cardiovascular death, HF, and MI/stroke), 14 overlapping differential metabolites in cardiovascular death and MI/stroke, 25 overlapping differential metabolites in cardiovascular death and HF, 15 differential metabolites specific to cardiovascular death, 56 differential metabolites specific to HF, and 9 differential metabolites specific to MI/stroke. **b** The alluvial diagram showed main classes of differential metabolites for individual cardiovascular events. **c** Upset plot illustrated the distinct metabolites among different cardiovascular events. The horizontal bar of differential metabolites set-size in the plot was represented in three colors, with blue for cardiovascular death (*n* = 94), red for HF (*n* = 121), and yellow for MI/stroke (*n* = 63). **d**–**f** Metabolic networks built with the specific key

combination of differential metabolites of individual events. The color of each node represented network cluster with the walktrap-algorithm. **d** Cardiovascular death; **e** Heart failure; **f** MI/stroke. **g** Dysregulated metabolic pathways linked with different cardiovascular events. The color of each point represented the type of metabolic pathway, the shape of points represented the dysregulated metabolic pathways of different cardiovascular events, with a circle representing cardiovascular death, a triangle representing heart failure, and a square representing MI/stroke. Multiple comparisons were adjusted for cardiovascular death and HF-related metabolic pathways. The tests used for pathway analyses were two-sided. CV cardiovascular; HF heart failure; hs-cTnT high sensitivity cardiac troponin T; LPA lysophosphatidic acid; LPC lysophosphatidylcholine; MI myocardial infarction; NT-proBNP N-terminal pro-brain natriuretic peptide; PC phosphatidylcholine; TIMI Thrombolysis In Myocardial Infarction.

---

cardiovascular events, among which the majority were middle and long-chain acylcarnitines. (2) The specific metabolites for individual cardiovascular events were identified, and in particular, glycerophospholipids were specific to HF development. (3) Middle and long-chain acylcarnitines served as the key metabolites in the metabolic network of cardiovascular death and MI/stroke, while PC and LPC played significant roles in the metabolic network of HF. (4) The pathway analyses revealed eight dysregulated metabolic pathways were shared among different cardiovascular outcomes, as well as six metabolic pathways specific to cardiovascular death, four metabolic pathways specific to HF, and two metabolic pathways specific to MI/stroke. (5) The inclusion of plasma metabolites improved the predictive capability of TIMI variables for the composite of cardiovascular events; adding the key metabolites combination to TIMI variables, hs-cTnT, and NT-proBNP significantly enhanced the predictive performance for HF risk.

To the best of our knowledge, this study is the first to provide the comprehensive evidence for the shared metabolic alteration among the composite of cardiovascular events, and specific dysregulated metabolic biomarkers and pathways that vary among different cardiovascular events in patients with CAD. These findings highlight the substantial heterogeneity in the pathological process of individual outcomes which were derived from CAD. Our study represents a starting point for the development of integrated prognostic metabolic tests with a two-step strategy to identify both shared and distinct metabolic risk factors for predicting different cardiovascular outcomes. This strategy involves utilizing the shared dysregulated metabolites (i.e., middle and long-chain acylcarnitines) to predict the overall risk of the composite of cardiovascular events, and further using the specific differential metabolites (i.e., PC and LPC) to predict HF risk. Although these hypotheses necessitate investigation using targeted metabolomics or other techniques, the two-step metabolic tests will trigger intensive monitoring patients with CAD, and furnish clinicians with the guidance aimed to tailor treatments effectively.

Our findings demonstrated that the middle and long-chain acylcarnitines comprise a significant portion of the dysregulated metabolites related to the composite of cardiovascular events, in comparison with controls. Carnitine dysmetabolism was also a shared pattern among different cardiovascular events. Previous studies have shown that elevated levels of even-chained acylcarnitines were identified to be independent predictors of adverse prognosis in patients with stable angina pectoris and with non-obstructive CAD[18,19]. In this study, we comprehensively evaluated the associations of acylcarnitines and cardiovascular events and discovered several novel metabolites, including dysregulated Car (16:2-O), Car (14:2), Car (16:1-O), Car (18:3-O), Car (16:3), Car (8:1-O), Car (14:1-O2), Car (16:3-O) and Car (6:1) in the composite of cardiovascular events group by using untargeted metabolomics. These middle (C6–C12) and long (C13–C20) chain acylcarnitines are mainly synthesized with the assistance of the carnitine acyltransferase system through the conjugation of L-carnitine and fatty acids, which serve as the primary energy source for the myocardium[20–22]. Acylcarnitines play a key role in transporting

acyl groups from the cytosol to the mitochondrial matrix, thereby enabling β-oxidation and the subsequent generation of essential energy for cellular activities[23]. Elevated levels of acylcarnitines may indicate impaired β-oxidation of fatty acids and altered mitochondrial metabolism[24]. Moreover, accumulated long-chain acylcarnitines can disrupt cardiac mechanical and electrophysiological processes via electrolyte disturbances and destabilization of the action potential and depolarization, even leading to death[25]. The levels of middle- and long-chain acylcarnitines are greater in the heart when fed compared to their levels in other tissues during fasting. Therefore, if the transfer of acylcarnitines from tissues to the bloodstream is influenced by their intracellular amounts, the heart could exhibit the maximum efflux of these acylcarnitines to the plasma. Makrecka-Kuka et. al found that the heart was the main contributor of blood long-chain acylcarnitines[26], which could be transported from cells across membranes into the circulation, even when there is no myocardial injury[26,27]. This suggests that acylcarnitines have the potential to be biomarkers for the early risk assessment of cardiovascular events, compared to hs-cTnT and NT-proBNP which could be detected in the blood after myocardial injury or dysfunction. However, further biological and population studies are necessary to fully understand and validate the clinical utility of acylcarnitines.

We found that TCA cycle was a markedly dysregulated metabolic pathway shared in patients with HF and MI/stroke. TCA cycle plays a crucial role in oxidative phosphorylation in the myocardiocytes. It is initiated by the reaction that combines acetyl-CoA, which is produced from the oxidation of fatty acids and pyruvate, or degradation of amino acids[28]. We also revealed that several amino acid related pathways, including cysteine and methionine, tyrosine, and tryptophan metabolism, were shared dysregulated metabolic pathways among the cardiovascular events. These dysregulated amino acid metabolisms were associated with pyruvate or acetyl-CoA which played crucial roles in TCA cycle. Disturbances of TCA cycle can cause mitochondrial dysfunction, resulting in oxidative stress and a pro-inflammatory state[28]. Previous study revealed that elevated levels of plasma isocitrate, a component of TCA cycle, were associated with a higher risk of cardiovascular events in patients with acute coronary syndrome (a subtype of CAD)[29]. The elevated levels of isocitrate indicated the inactivity of mitochondrial NADP+-isocitrate dehydrogenase, which was an enzyme responsible for the oxidative decarboxylation of isocitrate[30].

In addition to endogenous metabolites, some exogenous small molecules associated with cardiovascular events were also identified in this study. Phthalide is a plastic exposure and has been readily detected and reported in human serum, blood, and urine[31,32]. Erucic acid is also an environmental exposure and has been reported in human plasma, and breast milk[33,34]. Free erucic acid is transported in the blood mainly bound to subdomain IIIA (Sudlow site II) of human serum albumin[35]. The albumin-bound erucic acid is then transported into tissue cells via the circulatory system, whereas, erucic acid present in ultra low-density lipoprotein and very low-density lipoprotein is

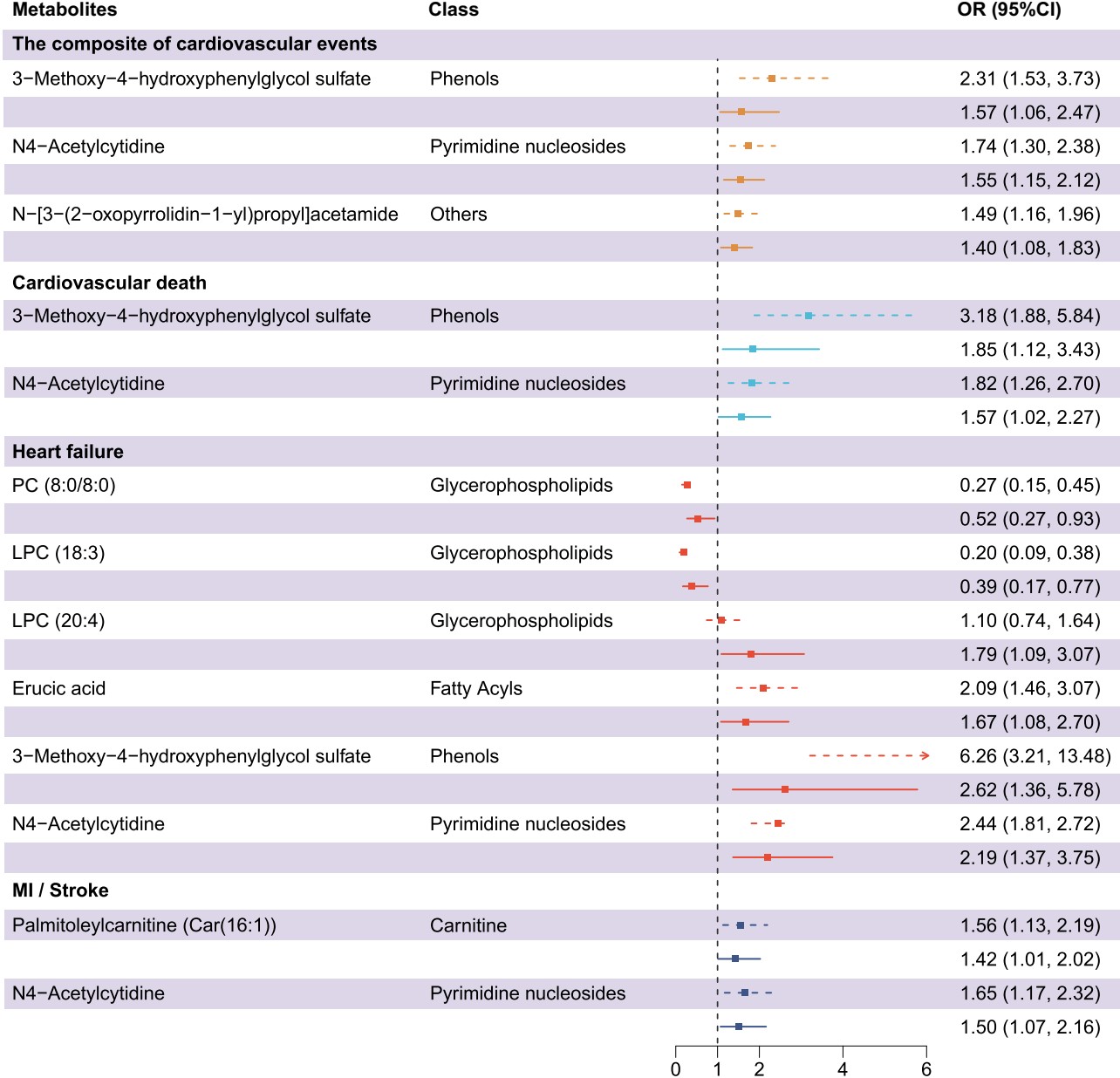

**Fig. 4 | The associations between differential metabolites and cardiovascular events.** The forest plot showed the ORs (95% CI) of differential metabolites with the composite ($n = 332$) and individual events (Cardiovascular death: $n = 242$; Heart failure: $n = 208$; MI/stroke: $n = 259$) in the validation set. Only differential metabolites both showed significant association with cardiovascular events in discovery set and validation set after adjusting for TIMI variables, hs-cTnT, and NT-proBNP were exhibited in the figure. The measure of centre for the error bars presented the ORs of differential metabolites, the dash line demonstrated the 95% CI of ORs after adjusting for TIMI variables, while the solid line showed the 95% CI of ORs after adjusting for TIMI variables, hs-cTnT, and NT-proBNP. hs-cTnT high sensitivity cardiac troponin T; LPC lysophosphatidylcholine; NT-proBNP N-terminal pro-brain natriuretic peptide; OR odds ratio; PC phosphatidylcholine; TIMI Thrombolysis In Myocardial Infarction.

broken down to free form with the aid of lipoprotein lipase in the endothelial cells and then passed on into cells[36]. Importantly, erucic acid has been showed to be associated with cardiotoxicity in both human and rat models[37]. A large-scale population study found an elevated plasma level of erucic acid was positively correlated with higher risk of congestive HF incidence, and these effects may result from the presence of other long-chain monounsaturated fatty acids[33].

We identified the distinct metabolic patterns and pathways that were specific for individual cardiovascular events. Specially, we found that glycerophospholipids alteration was specific in patients occurring with HF, suggesting that the metabolic mechanisms involved in the development of HF differ from other cardiovascular events. We observed that baseline levels of PC and LPC were decreased in patients

occurring with HF, resulting in disorders in glycerophospholipid metabolism which contributed to the development of HF. A previous lipidomic study showed that lipids were inhibited in patients with post-myocardial infarction heart failure, and PC (22:4/14:1) (AUC = 0.838) could be potential biomarkers for the event[38]. Another prospective HF case-control study found that ceramide (16:0), PC 32:0, and several lipidomics patterns were significantly associated with a higher risk of incident HF[39]. And lower levels of circulating PC (36:4) and LPC (18:2) were proved to be associated with higher risk of incident HF with reduced ejection fraction in the community[40]. These findings highlight the role of disturbed lipid metabolism in the pathophysiological processes of HF. Disorders of PC and disturbances in membrane phospholipid homeostasis can disturb myocardial metabolism and cellular

**Table 2 | Predictive value of plasma metabolites**

| Model | The composite of CV events | | CV death | | Heart failure | | MI/Stroke | |
|---|---|---|---|---|---|---|---|---|
| | AUC | p | AUC | p | AUC | p | AUC | p |
| The key metabolites combination | 0.64 (0.58, 0.70) [b] | – | 0.71 (0.64, 0.79) [c] | – | 0.89 (0.84, 0.94) [d] | – | 0.63 (0.55, 0.70) [e] | – |
| TIMI variables [a] | 0.63 (0.57, 0.69) | 0.010 | 0.68 (0.60, 0.74) | 0.008 | 0.68 (0.59, 0.77) | $1.483 \times 10^{-6}$ | 0.63 (0.56, 0.70) | 0.061 |
| TIMI variables and Metabolites | 0.70 (0.64, 0.75) | | 0.76 (0.70, 0.83) | | 0.91 (0.86, 0.94) | | 0.69 (0.62, 0.76) | |
| hs-cTnT | 0.67 (0.61, 0.72) | 0.543 | 0.73 (0.66, 0.80) | 0.879 | 0.82 (0.74, 0.88) | 0.029 | 0.63 (0.56, 0.70) | 0.924 |
| hs-cTnT and Metabolites | 0.65 (0.59, 0.70) | | 0.73 (0.66, 0.80) | | 0.89 (0.84, 0.94) | | 0.63 (0.55, 0.70) | |
| NT-proBNP | 0.70 (0.64, 0.76) | 0.697 | 0.80 (0.74, 0.86) | 0.497 | 0.88 (0.83, 0.93) | 0.003 | 0.61 (0.54, 0.69) | 0.232 |
| NT-proBNP and Metabolites | 0.71 (0.65, 0.76) | | 0.81 (0.75, 0.87) | | 0.94 (0.91, 0.97) | | 0.65 (0.58, 0.72) | |

The DeLong's test for two correlated ROC curves were two-sided test. Considering that only three tests were performed for each outcome, no multiple comparison correction was made for each test.
*CV* cardiovascular, *MI* myocardial infarction.
[a]TIMI variables included age, current smoking, hypertension, diabetes mellitus, previous stroke, previous HF, previous PAD, previous PCI/CABG, and eGFR.
[b]The key metabolites combination of the composite of cardiovascular events were identified by Lasso algorithm, including cystine, hexenoylcarnitine (Car(6:1)), Oxooctadecadienoylcarnitine (Car(18:3-O)), Hexadecatrienoylcarnitine (Car(16:3)), 5-Acetylamino–6-amino-3-methyluracil (AAMU), erucic acid, suberic acid, Vanillylmandelic acid, Mandelic acid, 3-(2,3,4-Trimethoxyphenyl) propanoic acid, N4-Acetylcytidine, N-[3-(2-oxopyrrolidin-1-yl)propyl]acetamide, 3-Hydroxyoctanoic acid, Phthalide.
[c]The key metabolites combination of cardiovascular death were identified by Lasso algorithm, including Tetradecadienoylcarnitine (Car(14:2)), Car(18:3-O), Hexacanedioic acid, Vanillylmandelic acid, N-[3-(2-oxopyrrolidin-1-yl)propyl]acetamide, 3-Hydroxyoctanoic acid, Phthalide, N,N-Dimethylarginine (ADMA), Homoarginine.
[d]The key metabolites combination of heart failure were identified by Lasso algorithm, including Cystine, N-Acetyl-arginine, Homoarginine, Erucic acid, N4-Acetylcytidine, 11beta-hydroxyandrost-4-ene-3,17-dione, Cortisol, LPA(20:5), LPC(20:4), LPC(O-16:0), PC(10:0/10:0), PC(22:2), PC(8:0/8:0).
[e]The key metabolites combination of myocardial infarction/stroke were identified by Lasso algorithm, including 1,7-Dimethylxanthine, 3-(2,3,4-Trimethoxyphenyl) propanoic acid, Erucic acid, Car(16:3), Mandelic acid, N4-Acetylcytidine, Oxooctanoylcarnitine (Car(8:1-O)), Palmitoleylcarnitine (Car(16:1)), Suberic acid, Theobromine, Theophylline, Vanillylmandelic acid.

signaling[41]. However, different PC or LPC species might have contrasting effects on future cardiovascular events. This may be influenced by the saturation (number of double bonds) and length of glycerophospholipids[42]. Generally, PC species with long-chain saturated or monounsaturated fatty acids have been found to be positively associated with mortality, but PC species containing long-chain polyunsaturated fatty acids may have a protective effect[43]. Further investigation is necessary to uncover the detailed underlying mechanisms for the varying effects of different PC or LPC species on cardiovascular events.

We found that plasma metabolites had incremental predictive capability in addition to TIMI variables, and/or NT-proBNP, hs-cTnT. The AUCs of TIMI variables for the composite and individual cardiovascular events were unsatisfactory in CAD patients (less than 0.70), consistent with previous literatures[17,44]. However, the inclusion of the combination of key metabolites improved the predictive capability of TIMI variables for the composite of cardiovascular events, cardiovascular death, and HF. Moreover, the addition of key metabolites can efficiently enhance the predictive performance for HF risk, compared to established risk factors, including TIMI variables, NT-proBNP, or hs-cTnT. These findings suggest that it is reasonable to develop a two-step strategy of the integrated prognostic metabolic tests for predicting the risk of the overall cardiovascular events and individual HF events.

The strengths of our study included: (1) The matched nested case-control design enabled us to minimize the influence of potential confounders on the analyses. (2) The study participants were divided into a discovery set and a validation set. All differential metabolite analyses were conducted in the discovery set, and the associations and predictive value of these metabolites were then further evaluated in the validation set, ensuring the reproducibility of our findings. (3) The untargeted metabolomic platform allowed us to depict a comprehensive view of metabolic deregulation in patients who experienced incident cardiovascular events. (4) We applied the same statistical workflow for the composite and individual cardiovascular events, enabling us to clarify the shared metabolic disturbance across different cardiovascular events and to characterize the metabolic alterations specific to each cardiovascular event.

Several limitations of this study needed to be considered. Firstly, the untargeted metabolomics approach used in this study was unable to obtain the absolute concentrations of metabolites. Employing a targeted metabolomics platform for metabolite quantification would strengthen these findings. Secondly, though we used a sufficient sample size in this study, further analysis using a larger prospective cohort would validate these associations between differential metabolites and cardiovascular events. Thirdly, although our findings are supported by previous studies, most of which were based on patient-level evidence. The underlying molecular mechanisms need to be investigated in animal studies. Fourthly, since all study participants were Chinese population, it is necessary to carefully evaluate the generalizability of these findings to other populations.

In summary, this untargeted metabolomics study clarified the shared and distinct metabolic alteration related to cardiovascular events in patients with CAD. Though the shared carnitine metabolic disturbance was related to the composite of cardiovascular events, glycerophospholipids were specific metabolites in HF development. The addition of metabolites efficiently improved the prediction performance of TIMI variables for the risk of overall cardiovascular events; and particularly, metabolites significantly enhanced the predictive capability of TIMI variables, hs-cTnT, and NT-proBNP for HF risk. These findings emphasized the potential value of metabolites identified by a two-step strategy on risk assessment of cardiovascular events and strengthened the understanding of the heterogenic mechanisms across different cardiovascular outcomes.

## Methods

### Participants and study design

This study was approved by the research ethics committee of Qilu Hospital of Shandong University, and accepted by Zibo Central Hospital. Written informed consent was obtained from each study participant. Patients from two participating hospitals of the BIPass cohort (Site 1: Qilu Hospital of Shandong University, Jinan, China; Site 2: Zibo Central Hospital, Zibo, China) were included in this study. The definitions and diagnostic criteria for CAD subtypes are presented in Supplementary Data 14. The inclusion and exclusion criteria of patients are provided in Supplementary Table 5. In this study, we conducted a nested case-control design, including 333 patients with incident cardiovascular events as case group and 333 patients without any events as control group (https://clinicaltrials.gov/, NCT05550805). The controls were randomly selected from all participants at risk, matched by age, sex, body mass index, current smoking, hypertension, diabetes,

and previous MI, according to a 1:1 propensity score matching at group level. In order to guarantee the reliability and repeatability of the selection of differential metabolites, we split patients into discovery and validation sets by a 1:1 ratio (334 patients in the discovery set, 332 patients in the validation set), and more details are provided in Fig. 1 and Supplementary Methods.

### Outcome assessment

The cardiovascular events were designated as cardiovascular death, HF, and MI/stroke. Both the composite and individual cardiovascular events were designated as outcome events. The detailed definitions of outcome events are provided in Supplementary Data 15. Outcome events were recorded at 12 months after hospital admission via follow-up telephone calls by trained research assistants at each participating site. Source medical documents were obtained for event adjudication by an independent clinical events committee, blinded to the metabolite measurement.

### Reagents and sample preparation for metabolomics

Frozen plasma samples were stored at −80 °C at the Shandong Provincial Clinical Research Center for Emergency and Critical Care Medicine, Jinan, China. Metabolomics analyses were performed centrally at Interdisciplinary Research Center on Biology and Chemistry, Shanghai Institute of Organic Chemistry, Chinese Academy of Sciences, by laboratory personnel who were blinded to the clinical outcomes.

Liquid chromatography tandem mass spectrometry (LC-MS) grade water and methanol (MeOH) were purchased from Honeywell (Muskegon, USA). Ammonium hydroxide ($NH_4OH$) and ammonium acetate ($NH_4OAc$) were purchased from Sigma-Aldrich (St. Louis, USA). Metabolite chemical standards were purchased from J&K (Beijing, China), Sigma (St. Louis, USA), Carbosynth (Berkshire, UK), TCI (Tokyo, Japan), and Energy Chemical (Shanghai, China).

Human plasma samples (50 μL) were extracted using 150 μL MeOH with internal standards (d3-leucine and d6-phenylalanine). The samples then vortexed for 30 s and sonicated for 15 min. To precipitate proteins, the samples were incubated for 1 h at −20 °C, followed by 15 min centrifugation at 13,500 rpm and 4 °C. The resulting supernatants were transferred to high-performance liquid chromatography (HPLC) vials and stored at −80 °C prior to LC-MS/MS analysis.

### Untargeted LC-MS analyses

The data acquisition was performed using an ultra-high-performance liquid chromatograph (UHPLC) system Thermo Scientific Vanquish UHPLC coupled to a Thermo Scientific Orbitrap Exploris 480. A Waters ACQUITY UPLC BEH Amide column (particle size, 1.7 μm; 10/0 mm (length) × 2.1 mm (i.d.)) and Kinetex C18 column (2.6 μm, 2.1 × 100 mm) were used for the LC separation and the column temperature was kept at 25 °C. For hydrophilic interaction liquid chromatography (HILIC) analysis, mobile phase A was 25 mM $NH_4OH$ + 25 mM $NH_4OAc$ in water, and B was ACN for both the positive (ESI+) and negative (ESI−) modes. The flow rate was 0.5 mL/min and the gradient was set as follows: 0–0.5 min, 95% B; 0.5–7 min, 95–65% B; 7–8 min, 65–40% B; 8–9 min, 40% B; 9–9.1 min, 40–95% B; 9.1–12 min, 95% B. The injection volume was 2 μL. For reverse-phase liquid chromatography (RPLC) analysis, mobile phase A was 0.01% acetic acid in water, and B was the mixture of IPA and ACN (1:1) for both the positive (ESI+) and negative (ESI−) modes. The flow rate was 0.3 mL/min and the gradient was set as follows: 0–1 min: 1% B; 1–8 min: 99% B; 8–9 min: 99% B; 9.0–9.1 min, 99–1% B; 9.1–12 min: 1% B; The injection volume was 2 μL. All the samples were randomly injected during data acquisition.

The data acquisition was operated in full MS-scan mode with positive/negative ion polarity switch for individual samples. The information-dependent acquisition (IDA) mode was used for quality control (QC) samples to acquire MS/MS spectra. The source parameters were set as follows: Spray voltage was set to 3000 V or −3000 V for positive or negative mode, respectively. The aux gas heater temperature was set as 350 °C. Sheath gas was set as 50 arb. Aux gas was set as 15 arb. The capillary temperature was set as 400 °C. The full MS resolution was set as 60,000 and the AGC target was 1e6 for positive or negative mode, respectively. Maximum IT was set as 100 ms. The mass range was set to 70–1200 Da. For the dd-MS2 settings, MS resolution was set as 30,000, and AGC target was set at 1e5. Maximum IT was set as 60 ms. The Top N setting was set as 6. Isolation width was set as 1.0 m/z Da. The MS/MS spectra of the QC sample were acquired under SNCE 20–30–40%. The dynamic exclusion was set as 3.0 s and the isotope exclusion was on.

### Metabolomics data processing

Raw MS data (.raw) files were transformed to the mzXML format by ProteoWizard (version 3.0.20360). Then we applied XCMS package (version 3.2; https://bioconductor.org/packages/release/bioc/html/xcms.html) to process mass spectrometry data for peak detection, retention time correction, and peak alignment[45]. Key parameters were set as follows: method, "centWave"; ppm, 10; snthr, 3; peakwidth, c (5,30); minfrac, 0.5. Metabolite annotation was performed using MetDNA (version 1.2.2; http://metdna.zhulab.cn/)[46,47]. The metabolite annotation parameters were set as "HILIC" or "RP" according to liquid chromatography mode, and "30" or "SNCE_20_30_40%" for collision energy. A total of 492 metabolites with level 1 identification were used for subsequent analyses. According to the definition of metabolomics standards initiative (MSI), level 1 means metabolites annotated through matching of accurate precursor m/z (MS1), retention time (RT), and tandem MS/MS spectra (MS2) with the standard metabolite library. Information on identified metabolites in details are provided in Supplementary Data 16. For each metabolite, outliers were set as missing values according to the intensity more than 5 standard deviation (SD), and then K-nearest neighbor (KNN) algorithm was applied to missing values imputation. All metabolites were scaled to standard normal distribution (mean = 0, SD = 1) for statistical analyses (Supplementary Fig. 15).

### Statistical analyses

Partial least-squares discriminant analysis (PLS-DA) was utilized to examine the disparities in metabolic profiles between the composite of cardiovascular events group and controls. For the differential metabolites analyses, the initial step involved employing the Wilcoxon rank test and fold change level. Except for MI/stroke ($p < 0.05$ and |log1.2(Fold change)| > 1), the threshold criteria of statistical significance were set as false discovery rate (FDR) $q$ value < 0.05 and |log1.2(Fold change)| > 1 for the composite and individual cardiovascular events. Powers of the Wilcoxon–Mann–Whitney test were evaluated by the G*Power software (version 3.1.9.7) and a t-test with two tails specific for the Wilcoxon–Mann-Whitney test (two groups) was used[48]. With the α level set at 0.05 and the effect size (d) of 0.5, the statistical power for the composite of cardiovascular events (control 167 vs case 167), cardiovascular death (control 167 vs case 82), HF (control 167 vs case 48), and MI/stroke (control 167 vs case 72) was found to be 0.994, 0.950, 0.844, and 0.932, respectively. Based on these calculations, the current sample size is sufficient to support the discovery of differential metabolic biomarkers (Supplementary Table 6).

To account for the independent associations of the metabolites with cardiovascular events, we conducted two logistic regression analyses adjusting for established risk factors. In Model 1, we adjusted for variables in TIMI score, including age, current smoking, hypertension, diabetes mellitus, previous stroke, previous HF, previous peripheral arterial disease, previous percutaneous coronary intervention/coronary artery bypass grafting, and eGFR; In model 2, we further

adjusted for NT-proBNP and hs-cTnT in addition to TIMI variables. Moreover, we ascertained the classification of these metabolites by cross-referencing each metabolite with the Human Metabolome Database (HMDB, https://hmdb.ca/).

To ensure the efficiency and brevity of the prediction model, we employed the least absolute shrinkage and selection operator (LASSO) algorithm to identify the combination of key differential metabolites in the prediction model (the combination of key metabolites). This algorithm was applied to those metabolites that demonstrated significant associations after adjusting for TIMI variables, NT-proBNP, and hs-cTnT in discovery set. Then, the combination of shared key metabolites was used for predicting the risk of the overall cardiovascular events, and the combination of specific key metabolites was adopted to predict the risk of individual cardiovascular events. Accordingly, prediction models were constructed using logistic regression analyses, including models based on the combination of key metabolites alone and models integrating key metabolites with TIMI variables, hs-cTnT and NT-proBNP, respectively. Model performance was assessed using metrics of discrimination (AUC) and calibration (Calibration curves and Hosmer–Lemeshow goodness-of-fit [GOF] statistic)[49,50].

We employed the conditional independence-based PC algorithm to construct metabolic network for individual cardiovascular events based on the combination of key metabolites[51]. To identify clusters within these metabolic networks, we applied the walk trap-algorithm using the igraph R-package (https://igraph.org/). We conducted pathway enrichment analyses using MetaboAnalyst 6.0 (https://www.metaboanalyst.ca/), which incorporated the Kyoto Encyclopedia of Genes and Genomes (KEGG) pathway database (accessed in December 2023)[52].

Differential metabolites analyses, metabolomics network analyses and pathway enrichment analyses were performed in the discovery set, while the prediction models were constructed in the discovery set and subsequently evaluated in the validation set. Statistical tests were 2-sided and the cut-off value of statistical significance was set at 0.05. All analyses were conducted using the R platform (version 3.6.0).

### Reporting summary
Further information on research design is available in the Nature Portfolio Reporting Summary linked to this article.

## Data availability
The data supporting the findings from this study are available within the manuscript and its supplementary information. The raw LC–MS data files generated in this study have been deposited in the National Omics Data Encyclopedia under accession code OEP005299 and are publicly available. Source data are provided with this paper. The raw individual participant data are protected and are not available due to data privacy laws. All the deidentified participant data for this study will be shared upon reasonable request by the corresponding author Yuguo Chen (chen919085@sdu.edu.cn). Requests will be responded to within 14 business days. The reuse of the deidentified participant data is subject to restrictions outlined in data use agreements, including limitations on data sharing and required confidentiality. Source data are provided with this paper.

## Code availability
Differential metabolomics analyses related codes in this study are available on the Code Ocean website (https://codeocean.com/capsule/8925883/tree).

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

## Acknowledgements

We thank the staff and participants of the BIPass cohort study for their valuable contributions. This study was supported by the National Key R&D Program of China (2017YFC0908700 (Y.Chen), 2020YFC1512700 (Y.Chen), 2022YFC3400700 (Z.J.Z)); Key Research and Development Program of Shandong Province (2021ZLGX02 (Y.Chen), 2021SFGC0503 (J.P)); Taishan Pandeng Scholar Program of Shandong Province (tspd20181220 (Y.Chen)); Taishan Young Scholar Program of Shandong Province (tsqn202306349 (C.P)); National Natural Science Foundation of China (82222064 (T.Z), 81973147 (T.Z), 22022411 (Z.J.Z), 82272240 (C.P)); Shanghai Municipal Science and Technology Major Project (2019SHZDZX02 (Z.J.Z)); Shanghai Key Laboratory of Aging Studies (19DZ2260400 (Z.J.Z)); and Shanghai Municipal Science and Technology Major Project (CEMS, CAS). Portions of Fig. 1 were created with BioRender.com.

## Author contributions

The principal investigator Y. Chen and the executive committee designed the study. Y. Chen, J.W., T.Z., Z.J.Z., J.Lv., and C.P. designed the present analyses. J.L. performed its statistical analyses and wrote the first draft of the manuscript. C.P. contributed to data visualization and wrote the first draft of the manuscript. Y. Cai, T.K., and F.R. contributed to the metabolomics sample preparation, data acquisition, data analysis, and manuscript writing. X.H. and C.W. contributed to the collection of plasma samples and statistical analysis. J.M., J.P., F.X., and S.W. contributed to the clinical information collection and outcome assessment. Z.J.Z. supervised the acquisition and analysis of metabolomics data. T.Z. supervised the data analysis of present analyses and reviewed and provided critical comments on drafts of the manuscript. J.W. supervised

the conduct of present analyses and provided critical revision of the manuscript for important content. Y. Chen reviewed and provided critical comments on drafts of the manuscript. All other authors participated in the data collection and literature search for this study and provided comments on drafts of the manuscript. J.L., C.P., and Y. Cai contributed equally to this work.

## Competing interests

The authors declare no competing interests.
