## [Peer Review File · Nature Communications]

REVIEWER COMMENTS

Reviewer #1 (Remarks to the Author):

The study combined metabolomic profiling of two sets of plasma samples from coronary artery disease patients (334 patients in the discovery set, 332 patients in the validation set) and bioinformatic analysis, leading to the identification of general (shared) and specific metabolite fingerprints in these patients. My review on the metabolomic analysis in this study observed some strength on the coverage of metabolomic platform, but also generated some concerns on the quality of metabolite identification.

---As stated in the Methods section (Line 372-375), the metabolites were identified using the level 1 criteria of the metabolomics standards initiative (MSI) by matching of accurate precursor m/z (MS1), retention time (RT), and tandem MS/MS spectra (MS2) with the standard metabolite library. I am not sure about the metabolite library in this case as many metabolite markers listed in this manuscript have uncommon structures and, in my knowledge, have not (or rarely) been reported in human samples. For example, almost all identified acylcarnitine markers (listed as Car(16:2-O), Car(14:2), Car(16:1-O), Car(18:3-O), Car(16:3), Car(8:1-O), Car(14:1-O2), Car(16:3-O) and Car(6:1)), contain uncommon (maybe unnatural) fatty acids. It is unclear how they were determined, what kind of metabolite library was used, and whether they have been identified previously in human blood. If their identities are correct, what are the sources of these acylcarnitines in human (which is not discussed in this manuscript). The same concern also applies to other metabolites, such as phthalide (a synthetic chemical), erucic acid (a plant fatty acid, not produced by human), and also D-glutamate and D-glutamine, two microbial amino acids (how do you know they are not L-glutamate and L-glutamine, which are the dominant forms in human body).

---What are the RT (retention time) values in supplemental tables? They are not consistent with the 12-min LC run in Supplemental Methods.

---These concerns compromise the values of the results and discussion, and may invalidate the discussion in this study.

Reviewer #2 (Remarks to the Author):

In this nested case-control study, authors tried to examine the relation of metabolic disturbance to cardiovascular events in patients with coronary artery disease [(CAD), mainly consisting of acute coronary syndrome (ACS)] utilizing plasma metabolomics technique. Though the topic of the manuscript is interesting and the methodological analysis is appropriate, the study was mainly limited by study design flaws with small sample size and short-term follow-up outcomes (12 months).

Firstly, The definition of ACS should be prescribed more precisely. If authors defined ACS and control on coronary angiography findings and clinical phenotypes, how did you determine the criteria of both groups, such as the degree of coronary stenosis?

Moreover, it is uncommon that heart failure (HF) is used as endpoint of cardiovascular events in patient with CAD for outcome studies, especially for ones with ACS. What is the definition of HF? In assessing events, the medical treatment, the disease severity of coronary stenosis, and different clinical phenotypes should be considered. Besides, significance of metabolomics should also be checked in subgroup analysis such as myocardial infarction vs stable CAD, stable CAD vs unstable CAD etc.

Furthermore, the factors which have secondary prevention effect for CAD, especially for ACS (e.g. β -blocker use and aspirin use et al.) need to be included for follow-up period, even utilizing a propensity score matched observation and parameters concerning cardiac function should systemically be shown in table 1.

Finally, the conclusion "These findings highlight the potential utility of plasma metabolites in early and tailored risk prediction in CAD patients" of this study did not exactly reflect the content of the study.

Reviewer #3 (Remarks to the Author):

Comments for the Author

The authors sought to identify novel prognostic metabolic biomarkers by considering metabolic disturbance associated with the composite outcome of combined and individual cardiovascular events. This reviewer has several concerns.

- 1) It seems that it defeats the purpose for developing models for the composite endpoint of death, heart failure and MI/stroke then model each individual event. What is the clinical rationale for that? This make the events much smaller
- 2) Sample size justification was not provided for this study. Why were 333 patients and 333 controls selected for this study?
- 3) It seems that the p-values in the tables are p-value and not FDR. However, it is confusing as reported in the results section
- 4) There is a huge focus on reporting p-values without prior hypotheses. The authors need to focus on the point estimates and confidence intervals and emphasize the clinical implication of their findings. Over 100 p-values have been reported! The p-values reported in Tables 1 and 2, and S1-S17 are not needed at all.

NEJM has recently released updated guidelines for reporting statistical results:

<https://www.nejm.org/doi/full/10.1056/NEJMe1906559>

- 5) There is no calibration for the models. Can the authors report that?
- 6) What is the utility of these models? If the purpose is to use in the clinic, will it be available online, so that clinicians can use it?

Reviewer #1 (Remarks to the Author):

General comment: *“The study combined metabolomic profiling of two sets of plasma samples from coronary artery disease patients (334 patients in the discovery set, 332 patients in the validation set) and bioinformatic analysis, leading to the identification of general (shared) and specific metabolite fingerprints in these patients. My review on the metabolomic analysis in this study observed some strength on the coverage of metabolomic platform, but also generated some concerns on the quality of metabolite identification.”*

Ans: We thank for the reviewer’s comments. We used the MetDNA software tool (Shen et al., Nat Commun, 2019, DOI: 10.1038/s41467-019-09550-x; Zhou et al., Nat Commun, 2022, DOI: 10.1038/s41467-022-34537-6; <http://metdna.zhulab.cn/>) for metabolite identification in this study. Metabolites identified in this study are level 1 identifications by matching of MS1, RT and MS/MS spectra with the metabolite library, and are used for discovery of metabolic disturbance associated with cardiovascular events. In the revised manuscript, we have provided information on the identified metabolites in details in **Supplementary data 1**. For individual comments raised by the reviewer, we have addressed them point-by-point as below.

Comment #1-1: *“---As stated in the Methods section (Line 372-375), the metabolites were identified using the level 1 criteria of the metabolomics standards initiative (MSI) by matching of accurate precursor m/z (MS1), retention time (RT), and tandem MS/MS spectra (MS2) with the standard metabolite library. I am not sure about the metabolite library in this case as many metabolite markers listed in this manuscript have uncommon structures and, in my knowledge, have not (or rarely) been reported in human samples. For example, almost all identified acylcarnitine markers (listed as Car(16:2-O), Car(14:2), Car(16:1-O), Car(18:3-O), Car(16:3), Car(8:1-O), Car(14:1-O2), Car(16:3-O) and Car(6:1)), contain uncommon (maybe unnatural) fatty acids. It is unclear how they were determined, what kind of metabolite library was used, and whether they have been identified previously in human blood. If their identities are correct, what are the sources of these acylcarnitines in human (which is not discussed in this manuscript).”*

Ans: We thank for the reviewer’s comments. We used the MetDNA software tool that we developed

to annotate metabolites in untargeted metabolomics data (Shen et al., Nat Commun, 2019, DOI: 10.1038/s41467-019-09550-x; Zhou et al., Nat Commun, 2022, DOI: 10.1038/s41467-022-34537-6; <http://metdna.zhulab.cn/>). Level 1 confidence in MetDNA means that metabolites are annotated through matching of MS1, RT and MS/MS spectra with the metabolite library. In this study, we kept a total of 492 metabolites with level 1 confidence identified by MetDNA for discovery of metabolic disturbance associated with cardiovascular events. We have provided information on the identified metabolites in details in **Supplementary data 1** in the revised manuscript.

Specifically for acylcarnitines, we identified them through MS1 match (m/z error < 15 ppm; most errors were less than 2.5 ppm), MS/MS match (score > 0.8), and retention time match (error < 25 s) against the in-house metabolite library. The detailed identification information has been provided in **Supplementary data 1**. We have also provided the mirror plots for MS/MS spectral match with dot product scores >0.8 for acylcarnitines in **Supplementary Figure 16**. In addition, we confirmed their identifications by manual confirmation of characteristic fragments and neutral loss for acylcarnitines, such as fragment ions with m/z 60.0808 Da ($C_3H_{10}N^+$), 85.0283 Da ($C_4H_5O_2^+$), 144.1019 Da ($C_7H_{14}NO_2^+$), and neutral loss signals with m/z 59.0735 Da (C_3H_9N) and 161.1052 Da ($C_7H_{15}NO_3$) in experimental MS/MS spectra (**Supplementary Figure 17**). The in-house metabolite library of acylcarnitines were curated by referring to the previous studies (Yu et al., Anal Chem, 2018, DOI: 10.1021/acs.analchem.7b05471; Yan et al., Anal Chem, 2020, DOI: 10.1021/acs.analchem.0c00129; Kind et al., Nat Methods, 2013, DOI:10.1038/nmeth.2551; Zhang et al., J Pharm Anal, 2024, DOI: 10.1016/j.jpha.2023.10.004). In brief, chemical standards of acylcarnitines were first analyzed and used to define characteristic fragment ions and retention time rules. Then, accurate mass, retention time, and MS/MS spectral information were used to define acylcarnitine homologues in biological samples. The generated library contains a total of 162 acylcarnitines, each with defined MS1, MS/MS spectrum, and retention time.

Importantly, these acylcarnitines including Car(16:2-O), Car(14:2), Car(16:1-O), Car(18:3-O), Car(16:3), Car(8:1-O), Car(14:1-O₂), Car(16:3-O) and Car(6:1) have been detected and reported in human plasma or human urine samples (Yu et al., Anal Chem, 2018, DOI: 10.1021/acs.analchem.7b05471; Vissing et al., J Clin Endocrinol Metab, 2019, DOI: 10.1210/jc.2019-00721; Zhang et al., J Pharm Anal, 2024, DOI: 10.1016/j.jpha.2023.10.004). For

example, hydroxylated acylcarnitines have been showed as indicators of a variety of physiological conditions such as mitochondrial myopathy and diabetic myocardium (Vissing et al., J Clin Endocrinol Metab, 2019, DOI: 10.1210/jc.2019-00721; Su et al., Biochemistry, 2005, DOI: 10.1021/bi047773a). With advancement of metabolomics technologies, we believe that more previously uncharacterized acycarnitines will be discovered for functional studies.

Supplementary Figure 16. Mirror plots of MS/MS match for acylcarnitines.

Supplementary Figure 17. Annotation of characteristic fragment ions in mass spectra of acylcarnitines.

Comment #1-2: “The same concern also applies to other metabolites, such as phthalide (a synthetic chemical), erucic acid (a plant fatty acid, not produced by human), and also D-glutamate and D-glutamine, two microbial amino acids (how do you know they are not L-glutamate and L-glutamine, which are the dominant forms in human body). ”

Ans: We thank for the reviewer’s comments. Phthalide was identified through matching against the chemical standard, including MS1 match (m/z error=0.2 ppm), MS/MS match (score=0.99), and retention time match (error=5.6 s). Erucic acid was also identified through matching against the chemical standard, including MS1 match (m/z error= 0.5 ppm), MS/MS match (score=1), and retention time match (error=4.5 s). Both are level 1 identifications. We have provided information on identification in details in **Supplementary data 1 and Supplementary Figure 18** in the revised manuscript.

Supplementary Figure 18. Mirror plots for MS/MS spectral match for phthalide and erucic acid.

We agree with the reviewer that phthalide is a synthetic chemical and erucic acid is a plant fatty acid. Phthalide is a plastic exposure and has been readily detected and reported in human serum, blood, and urine (Wang et al., Healthcare (Basel), 2021, DOI: 10.3390/healthcare9050603; Eales et al., Environ Int, 2022, DOI: 10.1016/j.envint.2021.106903). Erucic acid is also an environmental exposures and has been reported in human plasma, and breast milk (Imamura et al., Circulation, 2013, DOI: 10.1161/CIRCULATIONAHA.112.001197; Duan et al., J Oleo Sci, 2024, DOI: 10.5650/jos.ess23146). Importantly, erucic acid has been showed to be associated with cardiotoxicity in both human and rat models (Galanty et al., Molecules, 2023, DOI: 10.3390/molecules28041924). Therefore, it is reasonable that phthalide and erucic acid were detected in human plasma in this study. We have followed the reviewer’s suggestion and have pointed out that phthalide and erucic acid are exogenous chemicals but not endogenous metabolites in human in the revised manuscript

(line 288-299, page 13).

D-glutamate and D-glutamine were NOT identified in this study. Instead, L-glutamate (HMDB0000148) and L-glutamine (HMDB0000641) were identified in this study. The issue of the presence of “D-glutamate and D-glutamine metabolism” originated from an ID number mapping error of MetaboAnalyst (version 5.0; accessed on June 29th, 2023) which was used for pathway enrichment analysis. We used HMDB0000148 (L-glutamate) as input for MetaboAnalyst, however, the ID C00302 (refer both D-glutamate and L-glutamate) in KEGG database was incorrectly mapped for subsequent pathway enrichment analysis. During the revision, we noticed that MetaboAnalyst was recently updated (version 6.0; accessed on March 1st, 2024) and this bug was resolved. Therefore, we re-performed the pathway enrichment analysis using the updated MetaboAnalyst and the same data generated in our study. At this time, the D-glutamate and D-glutamine metabolism pathway was not enriched. In the revised manuscript, we have updated the pathway enrichment result analyzed by the latest MetaboAnalyst (version 6.0) in Figure 3g and have added the version information (line 127-131, page 6; line 174-186, page 8; line 429, page 19). We thank for the reviewer to point this out.

Figure 3g (updated). Significantly dysregulated metabolic pathways linked with different cardiovascular events.

Comment #2: “---What are the RT (retention time) values in Supplementary tables? They are not consistent with the 12-min LC run in Supplementary Methods.”

Ans: We thank for the reviewer’s comment. The unit of RT (retention time) is in seconds in Supplementary tables. Therefore, they are consistent with the 12-min LC run (720 seconds). We have added the unit of retention time in the revised manuscript (**Supplementary Table 6-9**).

Reviewer #2:

Comment #1: *“In this nested case-control study, authors tried to examine the relation of metabolic disturbance to cardiovascular events in patients with coronary artery disease [(CAD), mainly consisting of acute coronary syndrome (ACS)] utilizing plasma metabolomics technique. Though the topic of the manuscript is interesting and the methodological analysis is appropriate, the study was mainly limited by study design flaws with small sample size and short-term follow-up outcomes (12 months).”*

Ans: We thank for the reviewer’s comment. In this study, we employed a nested case-control design to undertake untargeted metabolomics research, utilizing blood samples from participants within the BIPASS cohort study (Wang et al., ESC Heart Fail, 2023, DOI: 10.1002/ehf2.14484; Wang et al., Lancet Reg Health West Pac, 2022, DOI: 10.1016/j.lanwpc.2022.100479). During follow-up period of cohort, we identified 333 individuals who experienced adverse cardiovascular events. Utilizing a 1:1 propensity score matching approach, we matched the cases with 333 controls from the same cohort who had not experienced any adverse cardiovascular events, thereby having the entire study population for the untargeted metabolomics analysis. Then, we split patients into discovery and validation set by a 1:1 ratio (334 patients in the discovery set, and 332 patients in the validation set), in order to guarantee the reliability and repeatability of the selection of differential metabolites. Although our overall sample size is relatively small, it is sufficient to support the analysis of differential metabolites in a case-control study.

Furthermore, we utilized the G*Power software (version 3.1.9.7) to calculate the statistical power of the non-parametric Wilcoxon-Mann-Whitney test (for two groups), using our current sample size (Faul et al., Behav Res Methods, 2009, DOI: 10.3758/BRM.41.4.1149). With the α level set at 0.05 and the effect size (d) of 0.5, the statistical power for the composite of cardiovascular events (control 167 vs case 167), cardiovascular death (control 167 vs case 82), heart failure (control 167 vs case 48), and myocardial infarction/stroke (control 167 vs case 72) were found to be 0.994, 0.950, 0.844, and 0.932, respectively. Therefore, we believe that the sample size is adequately robust to support the conclusions of subsequent analyses. We have added the sample size calculation in the revised manuscript (line 398-405, page 18).

As for the 12-month follow-up period, previous studies have already demonstrated the necessity of evaluating cardiovascular events within 12 months for CAD patients (Stone et al., JAMA, 2007, DOI: 10.1001/jama.298.21.2497; Stone et al., Lancet, 2013, DOI: 10.1016/S0140-6736(13)61170-8; Kandzari et al., Circulation, 2017, DOI: 10.1161/CIRCULATIONAHA.117.028885). However, 12-month time point usually is regarded as a short-term follow-up. We appreciated that you pointed out this issue and have added this limitation in the revised manuscript (line 343-345, page 15). In response to your concerns and suggestions, we will continue to extend the follow-up period in the future and validate our findings in a longer follow-up time.

Comment #2: *“Firstly, The definition of ACS should be prescribed more precisely. If authors defined ACS and control on coronary angiography findings and clinical phenotypes, how did you determine the criteria of both groups, such as the degree of coronary stenosis?”*

Ans: We thank for the reviewer’s comment. We have followed the reviewer’s suggestion and provided the predefined definitions of ACS and stable angina (SA) in the revised manuscript (Supplementary Table 18). The definitions and diagnostic criteria in this study were mainly based on cardiac biomarkers (troponin), together with clinical symptoms and electrocardiograph (ECG), according to the previously published definitions and literatures as well as the consensus of research steering committee of the BIPass study (Thygesen et al., J Am Coll Cardiol, 2018, DOI: 10.1016/j.jacc.2018.08.1038; Cannon et al., J Am Coll Cardiol, 2013, DOI: 10.1016/j.jacc.2012.10.005; Wang et al., Lancet Reg Health West Pac, 2022, DOI: 10.1016/j.lanwpc.2022.100479).

Supplementary Table 18. The definitions and diagnostic criteria for stable angina and acute coronary syndromes.

Disease	Definition
Stable angina (SA)	Typical or atypical angina, without any changes in the frequency or severity of attacks over the past 6 weeks, lasting ≤ 10 minutes, can be relieved by rest and/or medication. The resting ECG may or may not exhibit ischemic changes, while functional imaging tests (exercise ECG and stress echocardiography) show ischemic alterations. The level of cardiac biomarkers (troponin) is ≤ the 99th percentile upper reference limit (URL).

	Note:  1) The patient may or may not have a history of coronary artery disease (CAD), myocardial infarction (MI), percutaneous coronary intervention (PCI) and coronary artery bypass grafting (CABG); 2) The Clinician considers the cause of angina to be myocardial ischemia, and excludes other diseases with similar symptoms, such as myocarditis, aortic dissection, lung cancer, pneumonia, etc. When there is no definite cause of angina or precordial discomfort, but myocardial ischemia cannot be ruled out, it is diagnosed as angina caused by myocardial ischemia, and further determination of whether it is SA should be based on the clinical characteristics; 3) Coronary angiography or coronary CTA showing no plaques is not used as a basis for excluding SA.
Unstable angina (UA)	One of the following criteria is necessary:  1) Angina that occurred at rest and was prolonged, usually lasting ≥ 10 min; 2) New-onset angina of at least Canadian Cardiovascular Society (CCS) classification III severity; 3) Recent acceleration of angina reflected by an increase in severity of at least 1 CCS class to at least CCS class III. The patient must also not have any biochemical evidence (cardiac troponin) of myocardial necrosis. Confirmed ischemic ECG changes is not necessary if the local cardiologists and an independent Eligibility Committee think a diagnosis of UA is established.
Myocardial infarction (MI)	Detection of a rise and/or fall of cardiac troponin concentrations with at least one value above the 99th percentile URL, together with at least one of the following:  1) Symptoms of acute myocardial ischemia; 2) New ischemic ECG changes [new ST-T changes or new left bundle branch block (LBBB)]; 3) Development of pathological Q waves in at least two contiguous leads; 4) Identification of thrombus in a major epicardial coronary (diameter ≥ 2.5 mm) by angiography including intracoronary imaging or by autopsy; 5) Imaging evidence of new loss of viable myocardium or new regional wall motion abnormality in a pattern consistent with an ischemic etiology. Note:  1) Relative changes of at least 20% compared with the previous level is necessary to judge a rise and/or fall of cardiac troponin; 2) ECG findings are integrated to classify MI into ST-segment elevation myocardial infarction (STEMI) or non-ST segment elevation myocardial infarction (NSTEMI).
Non-ST segment elevation myocardial infarction (NSTEMI)	The diagnosis of MI is established, together with the ECG findings:  1) New horizontal or downsloping ST-depression ≥ 0.5 mm in 2 contiguous leads and/or T inversion > 1 mm in two contiguous leads with prominent R wave or R/S ratio > 1; 2) No new ST-elevation seen on the serial ECG.

ST-segment elevation myocardial infarction (STEMI)	The diagnosis of MI is established, together with the one of the ECG findings: 1) New or presumed new ST-segment elevation at the J-point in 2 contiguous leads with the cut-point: ≥ 1mm in all leads other than leads V2-V3 where the following cut-points apply: ≥ 2 mm in men ≥ 40 years; ≥ 2.5 mm in men < 40 years, or ≥ 1.5 mm in women regardless of age; 2) New or presumed new LBBB.
---	---

ACS included UA and MI, with MI being further classified into NSTEMI and STEMI.

In our study, coronary angiography findings were not used to define ACS and SA. There were two reasons in the determination of diagnostic criteria for the enrollment of patients. First, it should be noted that not all CAD patients presenting with coronary stenosis. Previous studies demonstrated that 67.4%~83% patients with non-ST-segment elevation ACS showed coronary stenosis greater than or equal to 50% (Bhatt et al., JAMA, 2022, DOI: 10.1001/jama.2022.0358; Linde et al., J Am Coll Cardiol, 2019, DOI: 10.1016/j.jacc.2019.12.012; Johnston et al., Am J Cardiol, 2015, doi: 10.1016/j.amjcard.2015.03.006). Moreover, studies on coronary angiography showed that up to 50%~70% patients had no evidence of obstructive CAD, but had demonstrable ischemia including symptoms of myocardial ischemia, ischemic ECG changes, and impaired coronary microvascular or macrovascular dysfunction (Beltrame et al., BMJ, 2021, doi: 10.1136/bmj-2021-060602; Kunadian et al., Eur Heart J, 2020, doi: 10.1093/eurheartj/ehaa503). Second, not all hospitalized CAD patients will undergo coronary angiography in routine clinical practice. If we solely analyzed the data from patients who underwent coronary angiography, our findings might not be applicable to all hospitalized ACS and SA patients. Therefore, in order to fulfill our study objective, which is to investigate the shared and distinct prognostic metabolites associated with the composite and individual cardiovascular events in patients hospitalized with CAD, we defined ACS and SA based on cardiac biomarkers, clinical symptoms and ECG.

Comment #3-1: *“Moreover, it is uncommon that heart failure (HF) is used as endpoint of cardiovascular events in patient with CAD for outcome studies, especially for ones with ACS. What is the definition of HF?”*

Ans: We thank for the reviewer’s comment. This study incorporated heart failure (HF) as one endpoint, mainly due to two considerations. First, from the perspective of disease prevalence, there is

a rising incidence of HF after CAD, especially ACS, making HF being an outcome event worthy of attention (Bahit et al., JACC Heart Fail, 2018, DOI: 10.1016/j.jchf.2017.09.015; Gerber et al., JAMA Cardiol, 2016, DOI: 10.1001/jamacardio.2016.0074; Song et al., Circ Res, 2017, DOI: 10.1161/CIRCRESAHA.117.311049; De Filippo et al., Int J Cardiol, 2023, DOI: 10.1016/j.ijcard.2022.10.146). Second, from the perspective of pathological mechanisms, metabolism disorders may play a crucial role in the occurrence of HF among the various cardiovascular events after CAD (Stanley WC et al., Physiol Rev, 2005, DOI: 10.1152/physrev.00006.2004; Bertero E et al., Nat Rev Cardiol, 2018, DOI: 10.1038/s41569-018-0044-6; Noordali H et al., Pharmacol Ther, 2018, DOI: 10.1016/j.pharmthera.2017.08.001).

Therefore, in order to investigate the shared and distinct prognostic metabolites associated with both the composite and individual cardiovascular events, we included HF as an endpoint in our study, besides cardiovascular death and MI/stroke. In the results, we discovered that glycerophospholipids alteration was specific to HF, and the addition of metabolites to TIMI variables, hs-cTnT and NT-proBNP significantly enhanced the prediction of HF risk. These findings highly supported our hypotheses in designing this untargeted metabolomics study.

The definitions of HF, including acute heart failure during index hospitalization and hospitalization for heart failure, have been provided in **Supplementary Table 20**.

Supplementary Table 20. The definitions of end points.

Outcome	Definition
Acute heart failure during index hospitalization	Defined as an event that meets the following criteria: 1) Subject has clinical signs and/or symptoms of heart failure, including new or worsening dyspnea, orthopnea, paroxysmal nocturnal dyspnea, increasing fatigue, worsening functional capacity or activity intolerance, or signs and/or symptoms of volume overload, OR 2) Results in intravenous (e.g., diuretic or vasoactive therapy) or invasive (e.g., ultrafiltration, IABP, mechanical assistance) treatment for heart failure.
Hospitalization for heart failure	Defined as an event that meets the following criteria of either (A and B and C) or D: 1) Requires hospitalization with treatment in any inpatient unit or ward in the hospital for at least 24 hours, including emergency department stay, AND 2) Subject has clinical signs and/or symptoms of heart failure, including new or worsening dyspnea, orthopnea, paroxysmal nocturnal dyspnea, increasing fatigue, worsening functional capacity or activity intolerance, or signs and/or symptoms of

	volume overload, AND 3) Results in intravenous (e.g., diuretic or vasoactive therapy) or invasive (e.g., ultrafiltration, IABP, mechanical assistance) treatment for heart failure. For the purpose of the clinical investigational plan, overnight stays at nursing home facilities, physical rehab or extended care facilities, including hospice, do not meet the clinical investigational plan definition of hospitalization. All hospitalizations, including the index hospitalization for the MitraClip procedure, if complicated by acute worsening heart failure that would have prompted an admission to hospital for heart failure, AND requires intravenous or invasive treatment AND hospitalization is extended by 24 hours, as defined above, will also be considered a heart failure hospitalization. Elective heart failure “tune-ups” that occur following the MitraClip procedure and prolong hospitalization will not count as a heart failure hospitalization. OR 4) Subject arrives in emergency department with clinical presentation meeting the criteria of heart failure but dies in the emergency department before hospital admission. Patients admitted for an LVAD or heart transplant will also be considered to have had a heart failure hospitalization.
--	---

Heart failure included acute heart failure during index hospitalization and hospitalization for heart failure.

Comment #3-2: *“In assessing events, the medical treatment, the disease severity of coronary stenosis, and different clinical phenotypes should be considered.”*

Ans: We thank for the reviewer’s comment. According to the reviewer’s valuable suggestions, we have assessed the association between differential metabolites and cardiovascular events, with the adjustment of pre-hospital medical treatments (β -receptor blockers, ACEI/ARB, statins, and aspirin), the severity of coronary stenosis, and different clinical phenotypes (systolic blood pressure, heart rate, and admission diagnosis), beyond the initial covariates (TIMI variables, hs-cTnT, and NT-proBNP). Our findings revealed that, even after adjusting for these factors in addition to TIMI variables, hs-cTnT, and NT-proBNP, the associations between differential metabolites and cardiovascular events remained significant. Detailed information is presented in **Supplementary Table 14, line 203-206, page 9:**

Supplementary Table 14. The associations between differential metabolites and cardiovascular events.

Metabolite	OR (95%CI) for model 1*	OR (95%CI) for model 2 [†]
The composite of cardiovascular events		
Cystine	1.59 (1.15, 2.24)	1.59 (1.14, 2.26)
Tetradecenedioylcarnitine (Car(14:1-O2))	1.58 (1.10, 2.36)	1.59 (1.09, 2.45)

Oxooctanoylcarnitine (Car(8:1-O))	1.85 (1.28, 2.79)	1.76 (1.20, 2.72)
Tetradecadienoylcarnitine (Car(14:2))	1.56 (1.12, 2.25)	1.60 (1.14, 2.35)
Oxopalmitoylcarnitine (Car(16:1-O))	1.38 (1.02, 1.92)	1.37 (1.00, 1.94)
Hexenoylcarnitine (Car(6:1))	1.47 (1.10, 2.02)	1.44 (1.06, 2.00)
Oxohexadecadienoylcarnitine (Car(18:3-O))	1.43 (1.08, 1.94)	1.45 (1.08, 1.99)
Oxopalmitoleylcarnitine (Car(16:2-O))	1.44 (1.06, 2.00)	1.45 (1.05, 2.04)
Hexadecatrienoylcarnitine (Car(16:3))	1.43 (1.07, 1.95)	1.45 (1.07, 2.00)
Oxohexadecadienoylcarnitine (Car(16:3-O))	1.43 (1.06, 1.96)	1.42 (1.05, 1.98)
Traumatic acid	1.46 (1.07, 2.10)	1.40 (1.03, 2.00)
Erucic acid	1.40 (1.07, 1.92)	1.54 (1.13, 2.25)
Suberic acid	1.46 (1.11, 1.98)	1.51 (1.14, 2.07)
Hexadecanedioic acid	1.35 (1.02, 1.82)	1.30 (0.99, 1.75)
Vanillylmandelic acid	2.22 (1.38, 3.73)	2.10 (1.29, 3.57)
Mandelic acid	1.46 (1.06, 2.20)	1.50 (1.08, 2.28)
3-Methoxy-4-hydroxyphenylglycol sulfate	1.68 (1.12, 2.70)	1.56 (1.05, 2.51)
3-(2,3,4-Trimethoxyphenyl)propanoic acid	1.55 (1.13, 2.16)	1.42 (1.04, 1.99)
N4-Acetylcytidine	1.86 (1.27, 2.83)	1.77 (1.20, 2.73)
N-[3-(2-oxopyrrolidin-1-yl)propyl]acetamide	1.39 (1.03, 1.91)	1.39 (1.01, 1.93)
3-Hydroxyoctanoic acid	1.39 (1.06, 1.87)	1.44 (1.09, 1.95)
Phthalide	0.73 (0.55, 0.94)	0.73 (0.54, 0.96)
5-Acetylamino-6-amino-3-methyluracil (AAMU)	0.73 (0.54, 0.94)	0.74 (0.55, 0.96)
Cardiovascular death		
N,N-Dimethylarginine (ADMA)	1.56 (1.02, 2.46)	1.65 (1.06, 2.69)
Homoarginine	0.56 (0.37, 0.82)	0.56 (0.36, 0.84)
Oxooctanoylcarnitine (Car(8:1-O))	1.77 (1.09, 3.00)	1.79 (1.09, 3.08)
Tetradecadienoylcarnitine (Car(14:2))	1.54 (1.02, 2.41)	1.63 (1.05, 2.60)
Oxohexadecadienoylcarnitine (Car(18:3-O))	1.50 (1.06, 2.19)	1.60 (1.11, 2.39)
Octadecatetraenoylcarnitine (Car(18:4))	1.44 (1.04, 2.07)	1.49 (1.06, 2.15)
Hexadecanedioic acid	1.47 (1.04, 2.12)	1.44 (1.02, 2.08)
Vanillylmandelic acid	2.11 (1.23, 3.87)	2.07 (1.19, 3.90)
3-Methoxy-4-hydroxyphenylglycol sulfate	1.77 (1.10, 2.99)	1.78 (1.10, 3.10)
3-(2,3,4-Trimethoxyphenyl)propanoic acid	1.69 (1.11, 2.61)	1.55 (1.02, 2.43)
N4-Acetylcytidine	1.76 (1.11, 2.94)	1.90 (1.16, 3.32)
N-[3-(2-oxopyrrolidin-1-yl)propyl]acetamide	1.57 (1.08, 2.33)	1.65 (1.11, 2.50)
3-Hydroxyoctanoic acid	1.54 (1.08, 2.28)	1.62 (1.12, 2.39)
Phthalide	0.48 (0.28, 0.75)	0.46 (0.26, 0.74)
Heart failure		
Cystine	2.43 (1.40, 4.46)	2.48 (1.38, 4.77)

N-Acetyl-arginine	0.51 (0.30, 0.82)	0.50 (0.26, 0.88)
Homoarginine	0.50 (0.30, 0.79)	0.45 (0.25, 0.75)
Oxoocytanoylcarnitine (Car(8:1-O))	1.95 (1.09, 3.60)	1.75 (0.94, 3.45)
Erucic acid	1.75 (1.17, 2.70)	1.90 (1.17, 3.23)
LPA(20:5)	0.58 (0.37, 0.88)	0.57 (0.34, 0.92)
LPC(20:4)	0.62 (0.38, 0.98)	0.60 (0.34, 1.02)
LPC(O-16:0)	0.56 (0.33, 0.92)	0.54 (0.29, 0.93)
PC(10:0/10:0)	0.56 (0.32, 0.95)	0.62 (0.33, 1.12)
LPC(16:1)	0.60 (0.37, 0.94)	0.71 (0.41, 1.18)
LPC(17:1)	0.57 (0.33, 0.93)	0.67 (0.37, 1.16)
PC(18:0)	0.56 (0.33, 0.89)	0.62 (0.35, 1.05)
PC(22:2)	0.47 (0.23, 0.86)	0.43 (0.20, 0.85)
LPC(18:3)	0.56 (0.31, 0.95)	0.69 (0.35, 1.28)
PC(8:0/8:0)	0.43 (0.23, 0.77)	0.54 (0.27, 1.03)
3-Methoxy-4-hydroxyphenylglycol sulfate	1.97 (1.09, 3.77)	1.89 (1.02, 3.71)
N4-Acetylcytidine	2.32 (1.24, 4.65)	1.83 (0.99, 3.59)
3-Hydroxyauric acid	1.75 (1.11, 2.82)	1.79 (1.11, 3.00)
Formylmethionine	1.65 (1.02, 2.77)	1.74 (1.00, 3.03)
11beta-hydroxyandrost-4-ene-3,17-dione	1.62 (1.06, 2.56)	1.96 (1.20, 3.38)
Cortisol	1.71 (1.13, 2.68)	2.02 (1.25, 3.48)
MI/Stroke		
1,7-Dimethylxanthine	0.53 (0.30, 0.86)	0.51 (0.28, 0.85)
2-Hydroxyphenylacetic acid	1.46 (1.03, 2.22)	1.60 (1.11, 2.55)
3-(2,3,4-Trimethoxyphenyl)propanoic acid	1.76 (1.22, 2.63)	1.70 (1.17, 2.55)
4-Hydroxyphenylacetic acid	1.46 (1.03, 2.22)	1.60 (1.11, 2.55)
Erucic acid	1.41 (1.02, 2.03)	1.56 (1.07, 2.45)
Hexadecatetraenoylcarnitine (Car(16:4))	1.44 (1.01, 2.07)	1.54 (1.07, 2.27)
Hexadecatrienoylcarnitine (Car(16:3))	1.58 (1.11, 2.32)	1.81 (1.23, 2.73)
Hexadecenedioylcarnitine (Car(16:1-O2))	1.77 (1.13, 3.35)	1.93 (1.21, 3.72)
Hexenoylcarnitine (Car(6:1))	1.56 (1.09, 2.26)	1.64 (1.12, 2.45)
Lauroylcarnitine	1.53 (1.05, 2.30)	1.71 (1.14, 2.68)
Mandelic acid	1.50 (1.07, 2.35)	1.65 (1.14, 2.73)
N4-Acetylcytidine	2.05 (1.28, 3.49)	2.15 (1.31, 3.81)
Oxodecanoylcarnitine (Car(10:1-O))	1.49 (1.07, 2.11)	1.47 (1.03, 2.13)
Oxohexadecadienoylcarnitine (Car(16:3-O))	1.48 (1.01, 2.20)	1.54 (1.02, 2.34)
Oxohexadecadienoylcarnitine (Car(18:3-O))	1.55 (1.06, 2.30)	1.78 (1.18, 2.74)
Oxoocytanoylcarnitine (Car(8:1-O))	2.04 (1.33, 3.31)	2.17 (1.38, 3.64)
Oxopalmitoleylcarnitine (Car(16:2-O))	1.59 (1.10, 2.36)	1.75 (1.18, 2.66)

Oxopalmitoylcarnitine (Car(16:1-O))	1.53 (1.05, 2.31)	1.75 (1.16, 2.74)
Palmitoleylcarnitine (Car(16:1))	1.48 (1.02, 2.20)	1.74 (1.17, 2.67)
Suberic acid	1.52 (1.09, 2.18)	1.66 (1.18, 2.44)
Tetradecadienoylcarnitine (Car(14:2))	1.73 (1.18, 2.68)	2.03 (1.33, 3.29)
Tetradecenedioylcarnitine (Car(14:1-O2))	1.62 (1.09, 2.59)	1.84 (1.20, 3.05)
Tetradecenoylcarnitine (Car(14:1))	1.55 (1.06, 2.32)	1.80 (1.20, 2.80)
Theobromine	0.53 (0.30, 0.86)	0.51 (0.28, 0.85)
Theophylline	0.53 (0.30, 0.86)	0.51 (0.28, 0.85)
Traumatic acid	1.48 (1.05, 2.17)	1.52 (1.08, 2.23)
Vanillylmandelic acid	2.29 (1.28, 4.34)	2.46 (1.33, 4.85)

* Model 1 adjusted for TIMI variables, hs-cTnT and NT-proBNP;

† Model 2 adjusted for TIMI variables, hs-cTnT, NT-proBNP, pre-hospital medical treatments (β -receptor blockers, ACEI/ARB, statins, and aspirin), the severity of coronary stenosis, and different clinical phenotypes (systolic blood pressure, heart rate, and admission diagnosis).

Comment #3-3: “Besides, significance of metabolomics should also be checked in subgroup analysis such as myocardial infarction vs stable CAD, stable CAD vs unstable CAD etc.”

Ans: We thank for the reviewer’s comment. We have further performed the sensitivity analyses in patients with acute coronary syndrome to investigate the significance of metabolites. In this study, we included 52 patients diagnosed with stable angina. During the follow-up period, there were 17 new cases of the composite of cardiovascular events, 3 cardiovascular deaths, 0 heart failure, and 14 MI/stroke. Calculating odds ratios and constructing prediction models with a small number of samples might lead to underpowered conclusions. Therefore, we performed sensitivity analyses only including patients with acute coronary syndrome. These results were generally consistent with the findings in the entire population. We have added the results of sensitivity analyses in the revised manuscript (Supplementary Table 16-17; line 226-227, page 10).

Supplementary Table 16. The key metabolites combination in prediction models for cardiovascular events.

Metabolite	FC	OR (95% CI) *
The composite of cardiovascular events		
Cystine	1.205	1.59 (1.14, 2.26)
Hexenoylcarnitine (Car(6:1))	1.434	1.42 (1.04, 1.99)
Oxohexadecadienoylcarnitine (Car(18:3-O))	1.362	1.34 (1.00, 1.82)
Hexadecatrienoylcarnitine (Car(16:3))	1.334	1.35 (1.01, 1.86)
Erucic acid	1.299	1.42 (1.08, 1.99)
Suberic acid	1.219	1.45 (1.09, 1.98)
Vanillylmandelic acid	1.712	1.99 (1.24, 3.34)
Mandelic acid	1.696	1.45 (1.05, 2.20)
3-(2,3,4-Trimethoxyphenyl)propanoic acid	1.487	1.53 (1.11, 2.16)
N4-Acetylcytidine	1.353	1.68 (1.15, 2.57)
N-[3-(2-oxopyrrolidin-1-yl)propyl]acetamide	1.303	1.44 (1.05, 1.99)
3-Hydroxyoctanoic acid	1.286	1.44 (1.08, 2.00)
Phthalide	0.684	0.70 (0.52, 0.92)
5-Acetylamino-6-amino-3-methyluracil (AAMU)	0.584	0.73 (0.54, 0.96)
Cardiovascular death		
N,N-Dimethylarginine (ADMA)	1.204	1.50 (0.98, 2.37)
Homoarginine	0.770	0.57 (0.37, 0.86)
Tetradecadienoylcarnitine (Car(14:2))	1.680	1.47 (0.97, 2.30)
Oxohexadecadienoylcarnitine (Car(18:3-O))	1.536	1.37 (0.97, 2.00)
Hexadecanedioic acid	1.283	1.42 (1.00, 2.06)

Vanillylmandelic acid	1.956	2.08 (1.22, 3.84)
N-[3-(2-oxopyrrolidin-1-yl)propyl]acetamide	1.402	1.57 (1.08, 2.33)
3-Hydroxyoctanoic acid	1.370	1.58 (1.09, 2.38)
Phthalide	0.543	0.51 (0.30, 0.79)
Heart failure		
Cystine	1.313	2.36 (1.36, 4.30)
N-Acetyl-arginine	0.798	0.49 (0.29, 0.81)
Homoarginine	0.691	0.51 (0.30, 0.80)
Erucic acid	1.452	1.73 (1.16, 2.66)
LPA(20:5)	0.737	0.58 (0.37, 0.89)
LPC(20:4)	0.730	0.62 (0.38, 0.99)
LPC(O-16:0)	0.724	0.57 (0.33, 0.93)
PC(10:0/10:0)	0.712	0.57 (0.33, 0.96)
PC(22:2)	0.598	0.48 (0.24, 0.89)
PC(8:0/8:0)	0.535	0.44 (0.24, 0.78)
N4-Acetylcytidine	1.467	2.22 (1.20, 4.41)
11beta-hydroxyandrost-4-ene-3,17-dione	1.252	1.59 (1.03, 2.53)
Cortisol	1.256	1.74 (1.14, 2.78)
MI/Stroke		
1,7-Dimethylxanthine	0.543	0.55 (0.30, 0.89)
3-(2,3,4-Trimethoxyphenyl)propanoic acid	1.535	1.83 (1.24, 2.79)
Erucic acid	1.260	1.48 (1.06, 2.21)
Hexadecatrienoylcarnitine (Car(16:3))	1.354	1.56 (1.09, 2.32)
Mandelic acid	1.793	1.52 (1.08, 2.40)
N4-Acetylcytidine	1.317	1.94 (1.20, 3.34)
Oxoctanoylcarnitine (Car(8:1-O))	1.621	1.99 (1.27, 3.34)
Palmitoleylcarnitine (Car(16:1))	1.239	1.46 (0.99, 2.21)
Suberic acid	1.214	1.56 (1.11, 2.28)
Theobromine	0.543	0.55 (0.30, 0.89)
Theophylline	0.543	0.55 (0.30, 0.89)
Vanillylmandelic acid	1.571	1.99 (1.11, 3.80)

The above results only including patients with acute coronary syndrome at admission;

* The association between metabolites and overall cardiovascular events, with adjustment of TIMI variables, hs-cTnT and NT-proBNP;

FC = fold change.

Supplementary Table 17. Predictive value of plasma metabolites in patients with acute coronary syndrome.

Model	The composite of cardiovascular events		Cardiovascular death		Heart failure		MI/Stroke	
	AUC	P	AUC	P	AUC	P	AUC	P
The key metabolites combination	0.65 (0.58, 0.70)*	-	0.70 (0.63, 0.77)†	-	0.88 (0.82, 0.93)‡	-	0.63 (0.56, 0.70)§	-
TIMI variables	0.63 (0.57, 0.69)	0.019	0.68 (0.60, 0.75)	0.019	0.68 (0.58, 0.77)	<0.001	0.62 (0.54, 0.70)	0.070
TIMI variables & Metabolites	0.70 (0.64, 0.75)		0.75 (0.69, 0.82)		0.90 (0.85, 0.94)		0.68 (0.61, 0.75)	
hs-cTnT	0.67 (0.61, 0.74)	0.546	0.72 (0.65, 0.79)	0.953	0.80 (0.73, 0.87)	0.035	0.65 (0.57, 0.72)	0.733
hs-cTnT & Metabolites	0.65 (0.59, 0.72)		0.72 (0.65, 0.79)		0.88 (0.83, 0.93)		0.63 (0.56, 0.71)	
NT-proBNP	0.71 (0.65, 0.76)	0.606	0.79 (0.73, 0.85)	0.518	0.88 (0.82, 0.92)	0.004	0.63 (0.55, 0.70)	0.201
NT-proBNP & Metabolites	0.72 (0.66, 0.77)		0.80 (0.74, 0.86)		0.94 (0.90, 0.97)		0.66 (0.59, 0.73)	

The above results only including patients with acute coronary syndrome at admission;

* The key metabolites combination of the composite of cardiovascular events were identified by Lasso algorithm, including cystine, hexenoylcarnitine (Car(6:1)), Oxohexadecadienoylcarnitine (Car(18:3-O)), Hexadecatrienoylcarnitine (Car(16:3)), 5-Acetylamino-6-amino-3-methyluracil (AAMU), erucic acid, suberic acid, Vanillylmandelic acid, Mandelic acid, 3-(2,3,4-Trimethoxyphenyl)propanoic acid, N4-Acetylcytidine, N-[3-(2-oxopyrrolidin-1-yl)propyl]acetamide, 3-Hydroxyoctanoic acid, Phthalide;

† The key metabolites combination of cardiovascular death were identified by Lasso algorithm, including Tetradecadienoylcarnitine (Car(14:2)), Car(18:3-O), Hexadecanedioic acid, Vanillylmandelic acid, N-[3-(2-oxopyrrolidin-1-yl)propyl]acetamide, 3-Hydroxyoctanoic acid, Phthalide, N,N-Dimethylarginine (ADMA), Homoarginine;

‡ The key metabolites combination of heart failure were identified by Lasso algorithm, including Cystine, N-Acetyl-arginine, Homoarginine, Erucic acid, N4-Acetylcytidine, 11beta-hydroxyandrost-4-ene-3,17-dione, Cortisol, LPA(20:5), LPC(20:4), LPC(O-16:0), PC(10:0/10:0), PC(22:2), PC(8:0/8:0);

§ The key metabolites combination of myocardial infarction/stroke were identified by Lasso algorithm, including 1,7-Dimethylxanthine, 3-(2,3,4-Trimethoxyphenyl)propanoic acid, Erucic acid, Car(16:3), Mandelic acid, N4-Acetylcytidine, Oxooctanoylcarnitine (Car(8:1-O)), Palmitoleylcarnitine (Car(16:1)), Suberic acid, Theobromine, Theophylline, Vanillylmandelic acid;

|| TIMI variables included age, current smoking, hypertension, diabetes mellitus, previous stroke, previous HF, previous PAD, previous PCI/CABG, and eGFR.

In response to your concerns, we further presented the distribution of the key metabolites combination in the prediction model among different subgroups, including myocardial infarction, unstable angina, and stable angina, for cardiovascular events. The distribution of differential metabolites for specific events revealed consistent trends across different subgroups. These results indicate that the significance of metabolites is stable across different subgroups. Detailed information on the distribution of metabolites is shown in the revised manuscript (**Supplementary Figure 11-14; line 224-226, page 10**):

Supplementary Figure 11 (continued on next page). The distribution of the key metabolites combination for the composite of cardiovascular events across different disease subtypes.

(a) Myocardial infarction; (b) Unstable angina; (c) Stable angina.

Supplementary Figure 11 (continued). The distribution of the key metabolites combination for the composite of cardiovascular events across different disease subtypes.

(a) Myocardial infarction; (b) Unstable angina; (c) Stable angina.

Supplementary Figure 12. The distribution of the key metabolites combination for cardiovascular death across different disease subtypes.

(a) Myocardial infarction; (b) Unstable angina; (c) Stable angina.

Supplementary Figure 13 (continued on next page). The distribution of the key metabolites combination for heart failure across different disease subtypes.

(a) Myocardial infarction; (b) Unstable angina; (c) Stable angina.

Supplementary Figure 13 (continued). The distribution of the key metabolites combination for heart failure across different disease subtypes.

(a) Myocardial infarction; (b) Unstable angina; (c) Stable angina.

Note: Due to the limitation of sample size, after subdividing the study population, the stable angina subgroup did not report any incident heart failure throughout the follow-up period. Consequently, the figure only depicts the distribution of metabolites within the control group.

Supplementary Figure 14 (continued on next page). The distribution of the key metabolites combination for myocardial infarction/stroke across different disease subtypes.

(a) Myocardial infarction; (b) Unstable angina; (c) Stable angina.

Supplementary Figure 14 (continued). The distribution of the key metabolites combination for myocardial infarction/stroke across different disease subtypes.

(a) Myocardial infarction; (b) Unstable angina; (c) Stable angina.

Comment #4: *“Furthermore, the factors which have secondary prevention effect for CAD, especially for ACS (e.g. β -blocker use and aspirin use et al.) need to be included for follow-up period, even utilizing a propensity score matched observation and parameters concerning cardiac function should systemically be shown in table 1.”*

Ans: We thank for the reviewer’s comment. In the revised manuscript, we have provided detailed information on parameters concerning cardiac function (left ventricular ejection fractions, LVEF) and the main treatments mentioned in your comments that may affect secondary prevention of coronary artery disease (CAD), including the medications (β -receptor blockers, ACEI/ARB, statins, and aspirin) before admission and 30 days, 6 months, and 12 months after enrollment (**Table 1, Supplementary Table 1-4**).

Supplementary Table 4. Medications during follow-up period.

Medications	Discovery set					Validation set				
	Control (n=167)	Case (n=167)	CV death (n=82)	Heart failure (n=48)	MI/stroke (n=72)	Control (n=166)	Case (n=166)	CV death (n=76)	Heart failure (n=42)	MI/stroke (n=93)
30 days after discharge										
β receptor blockers, n (%)	127 (76.5)	125 (81.2)	54 (78.3)	39 (86.7)	58 (80.6)	119 (73.0)	101 (73.2)	33 (66.0)	18 (60.0)	67 (79.8)
ACEI/ARB, n (%)	87 (52.4)	89 (57.8)	38 (55.1)	26 (57.8)	42 (58.3)	89 (54.6)	74 (53.6)	33 (66.0)	18 (60.0)	39 (46.4)
Statins, n (%)	158 (95.2)	138 (89.6)	58 (84.1)	38 (84.4)	66 (91.7)	150 (92.0)	130 (94.2)	46 (92.0)	28 (93.3)	81 (96.4)
Aspirin, n (%)	155 (93.4)	140 (90.9)	61 (88.4)	41 (91.1)	65 (90.3)	145 (89.0)	131 (94.9)	45 (90.0)	29 (96.7)	83 (98.8)
6 months after discharge										
β receptor blockers, n (%)	117 (70.5)	104 (79.4)	34 (73.9)	33 (82.5)	51 (78.5)	116 (71.6)	92 (73.0)	25 (64.1)	15 (57.7)	61 (77.2)
ACEI/ARB, n (%)	72 (43.4)	57 (43.5)	20 (43.5)	18 (45.0)	26 (40.0)	82 (50.6)	62 (49.2)	23 (59.0)	17 (65.4)	36 (45.6)
Statins, n (%)	146 (88.0)	113 (86.3)	36 (78.3)	33 (82.5)	57 (87.7)	144 (88.9)	117 (92.9)	36 (92.3)	25 (96.2)	74 (93.7)
Aspirin, n (%)	147 (88.6)	113 (86.3)	36 (78.3)	33 (82.5)	56 (86.2)	141 (87.0)	120 (95.2)	36 (92.3)	25 (96.2)	77 (97.5)
12 months after discharge										
β receptor blockers, n (%)	113 (68.5)	92 (78.0)	24 (72.7)	31 (86.1)	48 (77.4)	113 (70.2)	88 (77.9)	21 (77.8)	14 (60.9)	60 (78.9)
ACEI/ARB, n (%)	67 (40.6)	47 (39.8)	12 (36.4)	18 (50.0)	24 (38.7)	75 (46.6)	52 (46.0)	18 (66.7)	11 (47.8)	33 (43.4)
Statins, n (%)	139 (84.2)	101 (85.6)	26 (78.8)	31 (86.1)	56 (90.3)	139 (86.3)	99 (87.6)	22 (81.5)	18 (78.3)	68 (89.5)
Aspirin, n (%)	143 (86.7)	99 (83.9)	26 (78.8)	30 (83.3)	51 (82.3)	135 (83.9)	105 (92.9)	25 (92.6)	21 (91.3)	72 (94.7)

Data are presented with n (%), Case indicates the composite of cardiovascular events.

ACEI = angiotensin converting enzyme inhibitors; ARB = angiotensin receptor blocker; CV = cardiovascular; MI = myocardial infarction.

Comment #5: *“Finally, the conclusion ”These findings highlight the potential utility of plasma metabolites in early and tailored risk prediction in CAD patients” of this study did not exactly reflect the content of the study.”*

Ans: We thank for the reviewer’s comment. We have rewritten the conclusion to enhance the clarity and precision of the study.

This study focused on identifying both shared and distinct metabolic alterations across different cardiovascular events (cardiovascular death, heart failure and MI/stroke) to discover potential metabolic biomarkers for clinical application, as well as investigating the metabolic heterogeneity associated with these events. The findings demonstrated the presence of both shared and unique metabolic changes among different cardiovascular events. Notably, alterations in the metabolism of middle and long chain acylcarnitines emerged as a common hallmark across the metabolic transitions in these events, with heart failure specifically characterized by modifications in glycerophospholipids. Furthermore, predictive models constructed from various metabolites combinations exhibited good discriminative and calibration capabilities for predicting adverse cardiovascular outcomes in patients with coronary artery disease.

Based on these findings, we concluded that **“This study highlights the potential significance of plasma metabolites on tailed risk assessment of cardiovascular events, and strengthens the understanding of the heterogenic mechanisms across different events.”** The new conclusion has been added in the revised Abstract section of the manuscript **(line 65-68, page 3)**.

Reviewer #3:

Comment #1: *“The authors sought to identify novel prognostic metabolic biomarkers by considering metabolic disturbance associated with the composite outcome of combined and individual cardiovascular events. This reviewer has several concerns.”*

Ans: We thank for the reviewer’s comment. We have carefully considered each of your concerns and have made substantial revisions to our manuscript to address these issues. Below, we respond to your concerns one by one.

Comment #2: *“It seems that it defeats the purpose for developing models for the composite endpoint of death, heart failure and MI/stroke then model each individual event. What is the clinical rationale for that? This make the events much smaller”*

Ans: We thank for the reviewer’s comment. The primary purpose of this study is to delineate the shared metabolic alterations associated with cardiovascular events in the CAD patients, and to further identify potential metabolic biomarkers related to different cardiovascular events. Therefore, we initially screened for shared differential metabolites across the composite of cardiovascular events, and subsequently disaggregating the composite to explore the distinct metabolic changes. Accordingly, we adopted the same rationale to construct predictive models for both the composite and individual events. We designed this untargeted metabolomics study by considering that the progression of future cardiovascular events in CAD constitutes multifaceted and complicated processes, with different cardiovascular events undergoing distinct metabolic and pathological mechanisms (Talmor-Barkan et al., Nat Med, 2022, DOI: 10.1038/s41591-022-01686-6; Oldgren et al., Eur Heart J, 2003, DOI: 10.1016/s0195-668x(02)00312-3). Previous studies were mainly focused on the metabolic changes associated with the composite of cardiovascular events, thereby neglecting the shared and unique metabolic transitions across different events. Emerging metabolomics technologies enable our hypothesis to be possible.

We acknowledge that dissecting composite events can indeed result in a lower number of occurrences for different outcomes. Thus, we have assessed the sample size during the process of screening differential metabolites in the revised manuscript. We utilized G*Power software (version 3.1.9.7) to calculate the statistical power of the non-parametric Wilcoxon-Mann-Whitney test (for

two groups) given our sample size (Faul et al., Behav Res Methods, 2009, DOI: 10.3758/BRM.41.4.1149). With the α level set at 0.05 and the effect size (d) of 0.5, the statistical power for the composite of cardiovascular events (control 167 vs case 167), cardiovascular death (control 167 vs case 82), heart failure (control 167 vs case 48), and myocardial infarction/stroke (control 167 vs case 72) were found to be 0.994, 0.950, 0.844, and 0.932, respectively. Based on these results, we believe that our current sample size is sufficient to support the discovery of differential metabolites. It is important to note that the analysis of prediction models in this study is primarily aimed to further validate the association between the identified potential metabolites and adverse cardiovascular events. Therefore, our sample size justification is specifically focused on the discovery of differential metabolites. We have revised the methods section of our manuscript accordingly (**line 398-405, page 18; Supplementary Table 21**).

Supplementary Table 21. Power of differential metabolites discovery.

Events	Number of controls	Number of cases	Power
The composite of cardiovascular events	167	167	0.994
Cardiovascular death	167	82	0.950
Heart failure	167	48	0.844
Myocardial infarction/Stroke	167	72	0.932

Comment #3: “Sample size justification was not provided for this study. Why were 333 patients and 333 controls selected for this study?”

Ans: We thank for the reviewer’s comment. We have followed the reviewer’s suggestion and have added the sample size justification within the Methods section of the revised manuscript (**line 398-405, page 18; Supplementary Table 21**). With the statistical power greater than 0.80, we concluded that the total sample size is adequately robust to support the conclusions of subsequent analyses.

In this study, we employed a nested case-control design to undertake untargeted metabolomics research, utilizing biological samples from participants within the BIPASS cohort study (Wang et al., ESC Heart Fail, 2023, DOI: 10.1002/ehf2.14484; Wang et al., Lancet Reg Health West Pac, 2022, DOI: 10.1016/j.lanwpc.2022.100479). During follow-up period, we identified 333 individuals who

experienced adverse cardiovascular events. Utilizing a 1:1 propensity score matching approach, these individuals were paired with 333 controls from the same cohort who had not experienced any adverse cardiovascular events, thereby constituting the entire study population for the untargeted metabolomics analysis (n=666).

Comment #4: *“It seems that the p-values in the tables are p-value and not FDR. However, it is confusing as reported in the results section.”*

Ans: We thank for the reviewer’s comment. We have followed the reviewer’s suggestion and revised the relevant tables in supplementary tables. In the revised manuscript, to avoid unnecessary misunderstandings, we have removed the *P*-values for association analyses in **Supplementary Tables 6-9**. In the results section of this study, the tables detailing the differential metabolites and pathway analysis include both *P*-values and false discovery rate (FDR) values, and marking them only in the footnotes may lead to misunderstandings. As you mentioned later regarding the updated guidelines for reporting statistical results (Harrington D et al., N Engl J Med, 2019, DOI: 10.1056/NEJMe1906559), it is preferable to replace *P*-values with association and 95% CI when neither the protocol nor the statistical analysis plan has specified methods used to adjust for multiplicity.

Comment #5: *“There is a huge focus on reporting p-values without prior hypotheses. The authors need to focus on the point estimates and confidence intervals and emphasize the clinical implication of their findings. Over 100 p-values have been reported! The p-values reported in Tables 1 and 2, and S1-S17 are not needed at all.”*

NEJM has recently released updated guidelines for reporting statistical results:

<https://www.nejm.org/doi/full/10.1056/NEJMe1906559>

Ans: We thank for the reviewer’s comment. We have carefully read the reference guidelines you provided, and have revised the reporting of statistical results in our manuscript (Harrington D et al., N Engl J Med, 2019, DOI: 10.1056/NEJMe1906559). We have placed emphasis on point estimates

and confidence intervals in the results section, removing the unnecessary *P*-values (Table 1, Supplementary Tables 1-3, Supplementary Tables 5-9, Supplementary Tables 11-14, line 111-121, page 5). We hope that these revisions could highlight the clinical implications of our findings for readers.

Comment #6: “*There is no calibration for the models. Can the authors report that?*”

Response: We thank for the reviewer’s comment. The calibration performance of the prediction models has been added in the revised manuscript as you suggested (line 220-223, page 10; line 423-425, page 19; Supplementary Table 15, Supplementary Figure 7-10). In this study, we assessed the calibration of these predictive models based on calibration curves and the Hosmer and Lemeshow goodness of fit (GOF) test (Hosmer et al., Stat Med, 1997, DOI: 10.1002/(sici)1097-0258(19970515)16:9<965::aid-sim509>3.0.co;2-o; Kundu et al., Eur J Epidemiol, 2011, DOI: 10.1007/s10654-011-9567-4). The results showed that most prediction models exhibited good calibration (*P*-values >0.05). In the predictive models containing hs-cTnT, we found that the calibration for predicting future adverse cardiovascular events based solely on hs-cTnT was poor (*P*-values <0.05). After incorporating metabolites into the model, the calibration of the predictive models significantly improved, suggesting the enhanced calibration of metabolites over hs-cTnT. The calibration performance of the prediction models was presented as follows:

Supplementary Table 15. The calibration performance of the prediction models.

Model	The composite of CV events		CV death		Heart failure		MI/Stroke	
	χ^2	P -value	χ^2	P -value	χ^2	P -value	χ^2	P -value
The key metabolites combination	12.235	0.141	8.239	0.411	5.038	0.754	5.993	0.648
TIMI variables	7.417	0.492	9.861	0.275	3.679	0.885	6.693	0.570
TIMI variables & Metabolites	7.993	0.434	13.697	0.090	5.987	0.649	3.018	0.933
hs-cTnT	25.098	0.001	23.166	0.003	33.120	<0.001	20.823	0.008
hs-cTnT & Metabolites	16.132	0.041	6.540	0.587	5.141	0.742	4.526	0.807
NT-proBNP	4.470	0.812	13.078	0.109	3.341	0.852	7.866	0.447
NT-proBNP & Metabolites	8.421	0.393	4.215	0.837	1.664	0.990	0.902	0.999

Supplementary Figure 7. The calibration curves of prediction models for the composite of cardiovascular events.

(a) The key metabolites combination; (b) NT-proBNP; (c) NT-proBNP & Metabolites; (d) hs-cTnT; (e) hs-cTnT & Metabolites; (f) TIMI variables; (g) TIMI variables & Metabolites.

Supplementary Figure 8. The calibration curves of prediction models for cardiovascular death.

(a) The key metabolites combination; (b) NT-proBNP; (c) NT-proBNP & Metabolites; (d) hs-cTnT; (e) hs-cTnT & Metabolites; (f) TIMI variables; (g) TIMI variables & Metabolites.

Supplementary Figure 9. The calibration curves of prediction models for heart failure.

(a) The key metabolites combination; (b) NT-proBNP; (c) NT-proBNP & Metabolites; (d) hs-cTnT; (e) hs-cTnT & Metabolites; (f) TIMI variables; (g) TIMI variables & Metabolites.

Supplementary Figure 10. The calibration curves of prediction models for myocardial infarction/stroke.

(a) The key metabolites combination; (b) NT-proBNP; (c) NT-proBNP & Metabolites; (d) hs-cTnT; (e) hs-cTnT & Metabolites; (f) TIMI variables; (g) TIMI variables & Metabolites.

Comment #7: *“What is the utility of these models? If the purpose is to use in the clinic, will it be available online, so that clinicians can use it?”*

Ans: We thank for the reviewer’s comment. The primary aim of this metabolomics study is to identify potential metabolic biomarkers associated with future cardiovascular events in CAD patients. To achieve this purpose, our study employed an untargeted metabolomics approach for the quantitative analysis of metabolites in plasma. Untargeted metabolomics facilitates a comprehensive and systematic examination of metabolites within human biological samples, enabling the identification of novel biomarkers (Johnson et al., Nat Rev Mol Cell Biol, 2016, DOI: 10.1038/nrm.2016.25). However, this untargeted metabolomics methodology primarily allows for the relative quantification of metabolites, which poses challenges for the direct application of the generated metabolomics data in the development and validation of clinical prediction models (Schrimpe-Rutledge et al., J Am Soc Mass Spectrom, 2016, DOI: 10.1007/s13361-016-1469-y).

Despite these limitations, we constructed a prediction model in the latter section of the manuscript based on the identified metabolites. This analysis was mainly aimed at reinforcing the association between these metabolites and adverse cardiovascular events. The prediction model demonstrated remarkable discriminative power and calibration, underscoring the potential clinical utility of the identified metabolites in forecasting future adverse cardiovascular outcomes. Future efforts will focus on the large-scale, multicenter, targeted validation of these potential biomarkers, with the goal of developing predictive models for adverse cardiovascular events in CAD populations. This perspective has been mentioned in the limitations section of our manuscript (**line 342-345, page 15**). Furthermore, we have made the prediction model and associated code available on the journal's website (<https://codeocean.com/capsule/8925883/tree>), aiming to serve as a reference for researchers performing untargeted metabolomics data analyses (**line 441-443, page 20**).

REVIEWERS' COMMENTS

Reviewer #1 (Remarks to the Author):

The responses and the revisions have addressed my comments on metabolomics. Some discussion on the sources and biological meaning of those acylcarnitines containing uncommon fatty acids are desirable, but not mandatory.

Reviewer #2 (Remarks to the Author):

I enjoyed reading this revised article; the points made by this reviewer have been addressed.

Reviewer #2 (Remarks on code availability):

no further comments

Reviewer #3 (Remarks to the Author):

The authors have addressed almost all my concerns.

This reviewer still believes that there is an overemphasis on statistical significance and some of the p-values/FDR are hard to interpret. It might be best to just report the p-values on the composite endpoints

Reviewer #1 (Remarks to the Author):

Comment #1: *“The responses and the revisions have addressed my comments on metabolomics. Some discussion on the sources and biological meaning of those acylcarnitines containing uncommon fatty acids are desirable, but not mandatory.”*

Ans: We thank for the reviewer’s comments. In the revision, we have added the sources and biological meaning of those acylcarnitines in the discussion section of the revised manuscript (**line 265-270, page 12**). In this study, we identified several novel acylcarnitines, including dysregulated Car (16:2-O), Car (14:2), Car (16:1-O), Car (18:3-O), Car (16:3), Car (8:1-O), Car (14:1-O2), Car (16:3-O) and Car (6:1) in the composite of cardiovascular events. These middle (C6-C12) and long (C13-C20) chain acylcarnitines in the body are mainly synthesized with the assistance of the carnitine acyltransferase system through the conjugation of L-carnitine and fatty acids, which serve as the primary energy source for the myocardium. (Maija et al., Pharmacol Rev, 2022, doi: 10.1124/pharmrev.121.000408; Schulze et al., Circ Res, 2016, DOI: 10.1161/CIRCRESAHA.116.306842). Acylcarnitines play a key role in transporting acyl groups from the cytosol to the mitochondrial matrix, thereby enabling β -oxidation and the subsequent generation of essential energy for cellular activities (Indiveri C et al., Mol Aspects Med, 2011, 10.1016/j.mam.2011.10.008). Elevated levels of acylcarnitines may indicate impaired β -oxidation of fatty acids and altered mitochondrial metabolism. Increasing evidence suggests that changes in the levels of circulating long-chain acylcarnitines are associated with various cardiovascular diseases (Guasch-Ferre, M. et al., Am J Clin Nutr, 2016, DOI: 10.3945/ajcn.116.130492). We hope that this revision meets your expectations and enhances the manuscript, and we are committed to exploring this topic in subsequent studies. Thank you for enhancing the depth of our discussion with your comments.

Reviewer #2:

Comment #1: *“I enjoyed reading this revised article; the points made by this reviewer have been addressed.”*

Ans: We thank for the reviewer’s comments. We are pleased to see that the revisions have adequately addressed the points raised.

Reviewer #3:

Comment #1: *“The authors have addressed almost all my concerns.*

This reviewer still believes that there is an overemphasis on statistical significance and some of the p-values/FDR are hard to interpret. It might be best to just report the p-values on the composite endpoints.”

Ans: We thank for the reviewer’s comments. In response to your advice, we have minimized the focus on statistical significance (p-values and FDR) in the main text. Specifically, the statistical significance from the Wilcoxon rank-sum test, which serve as the basis for our differential metabolite screening, have now been relegated to supplementary materials and are not discussed in the main text (**Supplementary Data 5-8**). Additionally, in our association analyses, we have only provided confidence intervals, without mentioning p-values. We hope that these adjustments adequately address your concerns by reducing the emphasis on statistical significance and enhancing the focus on the clinical relevance of our findings. We appreciate your insights and hope that the explanations provided clarify the focus and intentions of our revisions.